# Position- and scale-invariant object-centered spatial localization in monkey frontoparietal cortex dynamically adapts to cognitive demand

Bahareh Taghizadeh [1,2], Ole Fortmann[1,3] & Alexander Gail [1,3,4,5]

Egocentric encoding is a well-known property of brain areas along the dorsal pathway. Different to previous experiments, which typically only demanded egocentric spatial processing during movement preparation, we designed a task where two male rhesus monkeys memorized an on-the-object target position and then planned a reach to this position after the object re-occurred at variable location with potentially different size. We found allocentric (in addition to egocentric) encoding in the dorsal stream reach planning areas, parietal reach region and dorsal premotor cortex, which is invariant with respect to the position, and, remarkably, also the size of the object. The dynamic adjustment from predominantly allocentric encoding during visual memory to predominantly egocentric during reach planning in the same brain areas and often the same neurons, suggests that the prevailing frame of reference is less a question of brain area or processing stream, but more of the cognitive demands.

Allocentric spatial cognition allows subjects to assess space independent of their own perspective. It is a fundamental skill supporting navigation[1], spatial judgment[2–5], and goal-directed movement behavior[6,7]. Allocentric encoding requires neural computation since the primary spatial sensory inputs are egocentric (body-relative) in nature, as the examples of visual retinotopy and tactile somatotopy show. Yet, the roles of different brain regions, especially of ventral versus dorsal stream processing, and neurocomputational mechanisms of allocentric encoding are still under debate[8–11]. Neurophysiology data at the single neuron level in the context of allocentric goal-directed reaching is lacking. Here, we ask if object-centered encoding can be observed during goal-directed reach planning in frontoparietal areas of the dorsal stream, i.e., areas which are mostly associated with egocentric frames of reference[12–14].

Allocentricity refers to viewpoint-invariant spatial encoding in a frame of reference that is relative to a location external to the

perceiver[8], including but not limited to object-, landmark- and world-centered encoding[15]. Here, we define object-centered encoding via a coordinate system that is anchored to an object and scales with its size. Under egocentricity, we subsume reference frames that are relative to the location or configuration of the subject's body, including direction of gaze (eye-centered), hand-, head-, or trunk(body)-centered.

Reaches are typically directed towards physical objects. Geometrical features such as shape and size of that object should be incorporated in movement planning to successfully direct the hand towards a suited part of the object. We may pick up a stick at different positions along its length, depending on the intended use. Such object-oriented reach goal localization may only depend on object properties and not on the spatial positioning of the object relative to the subject. This would mark a case of reach-associated, yet object-centered allocentric spatial processing for the localization of the target position of the reach. Additionally, the inherently egocentric changes in

[1]Sensorimotor Group, German Primate Center, Göttingen, Germany. [2] School of Cognitive Science, Institute for Research in Fundamental Sciences (IPM), P.O. Box 19395-5746 Tehran, Iran. [3]Faculty of Biology and Psychology, University of Göttingen, Göttingen, Germany. [4]Bernstein Center for Computational Neuroscience, Göttingen, Germany. [5]Leibniz ScienceCampus Primate Cognition, Göttingen, Germany. ✉e-mail: agail@gwdg.de

body-configuration needed to accomplish the reach are relevant for planning and implementing it. This suggests that spatial processing needs to be cognitively controlled so that those spatial parameters are available at different stages of action preparation that are relevant in the respective moment. In the earlier phase, when deciding where along the stick to pick it up, object-centered encoding relative to the stick is most relevant. Later, for planning the physical movement of the hand, egocentric postural signals might be more relevant. It is an open question how dynamically changing allo- and egocentric spatial cognitive demands for planning goal-directed movements are fulfilled in the frontoparietal reach planning network.

Human neuroimaging studies on allocentricity in the context of perceptually judging the spatial locations of visual objects (with verbal or button-press responses) suggest an anatomical segregation, where the dorsal stream vision-for-action processing is predominantly ego-centric and the ventral stream perceptual and object-recognition processing encompasses also allocentric representations[3,16]. Yet, which processing predominates may be task dependent, and it has been shown that allocentric spatial judgments can activate networks comprising areas of both dorsal and ventral streams[5,17]. A recent electrophysiology study showed when monkeys were instructed to perceptually judge the direction of motion of a cloud of dots in head- or world-centered coordinates, neurons in ventral intraparietal area (VIP) changed their spatial reference frame depending on the task instruction in different trials[18]. While this points to the possibility of context-dependent allocentric encoding in the visual cortex, the perceptual task of judging object motion during self-motion is very different from aiming a reach towards an object-relative location in terms of spatial-cognitive demand.

Human neuroimaging studies on target localization partly also indicated activation of overlapping regions of the frontoparietal network for planning and guiding goal-directed reach and saccade movements relative to egocentric and allocentric spatial references[19–23]. However, the role of dorsal stream in allocentric processing of target location is not clear from imaging studies. While allocentric compared to egocentric encoding of reach target distance and direction led to higher blood-oxygen-level-dependent (BOLD) activity in dorsal premotor cortex (PMd) and right posterior intraparietal sulcus of human subjects[24], others did not find such preference along the dorsal stream[22,23].

Single neuron recordings in monkeys revealed how sensory information is transformed into motor goal information within the frontoparietal network during reach planning[25]. Neurons in monkey posterior medial intraparietal cortex (parietal reach region, PRR) and PMd are selective for the spatial location of reach goals[26–34]. In monkeys that aim their reach at visually instructed positions, while controlling gaze independently, different frames of reference have been described, all of which were egocentric. Many studies reported encoding of reach goals in predominantly gaze-centered, in PRR[12,35–39] and PMd[40], or predominantly hand-centered, in PRR[14,41,42] and PMd[41,43,44] encoding. Encoding within these areas is typically not exclusive but rather intermediate with mixed[45,46] and more complex selectivities[13], also found in humans[47].

The prevailing view of dorsal stream processing of spatial locations that emerged from these neurophysiological findings is that different visual and somatosensory inputs, which are of egocentric nature, become integrated in the parietal multisensory association cortex to compute reach-relevant spatial information in different egocentric reference frames. Different anatomical nodes were shown to have a predominance for reference frames being centered on different body parts (see above) and the result of such feedforward integration[10,11,48] is supposed to be fed to more motor-related areas in the frontal lobe. Yet, if also allocentric spatial cognitive processing in the context of goal-directed reaching exists in parietal cortex is unclear, since single-cell electrophysiology studies directly comparing egocentric and allocentric reference frames during reach planning do not exist.

Here we directly compare posterior medial intraparietal cortex (parietal reach region, PRR) and dorsal premotor cortex (PMd) at the single neuron level in rhesus monkeys in an object-centered allocentric reach task that sequentially mandates spatial target memory and reach planning. Animals had to identify reach goals relative to a visual object where the object could appear randomly at two potential positions on the screen. We asked if neural selectivity patterns are best explained as a function of the target location on the object (object-centered hypothesis), or the target location relative to the body of the animal (egocentric hypothesis). We tested both position- and size-invariant object-centered encoding against egocentric encoding. In contrast to the prevailing view, we report that neurons in PRR and PMd encode visual cues and reach targets in object-centered as well as egocentric reference frames, with the predominant reference frame in both areas being dynamically adjusted based on cognitive demands.

## Results

Two male rhesus monkeys (*Macaca mulatta*) were trained to perform memory guided reaches towards variable positions on an elongated visual object on a touchscreen. The object had variable position relative to the animal, allowing to dissociate object-centered (allocentric) locations from body-centered (egocentric) reach goal locations (Fig. 1). Two instructed delays allowed separately investigating spatial frames of reference during visual memory and reach planning, respectively. The animals memorized the location of a briefly flashed peripheral visual cue at one of five positions on an elongated visual object (Fig. 1a; reference object) to later reach towards this on-the-object position irrespective of object location on the screen (reach object). The egocentric cue positions varied independently of their object-centered positions since the reference object was presented with a left or right offset relative to the screen center, randomly in each trial (Fig. 1b). After a first delay period (visual memory), the object re-appeared as reach object, again randomly with a left or right offset. After a second delay (reach planning), the monkeys had to reach to the previously cued target position on the object. The location of the reference object was not predictive for the location of the reach object. In experiment I (Exp-I; Fig. 1b), reference and reach object were visually identical but their location could be shifted horizontally to test for position-invariant ( = object-centered) encoding. In experiment II (Exp-II, Fig. 1c), additionally, the size could vary between reference and reach objects to test for position- and scale-invariant object-centered encoding (see Methods for details). In Exp-II, reference and reach object were horizontally and vertically offset to each other, such that they were always position-incongruent in both horizontal and vertical dimensions.

### Object-centered vs egocentric encoding hypothesis

In Exp-I, we asked if the fronto-parietal network encodes the location of the cue (visual memory) and the associated reach goal (reach planning) predominantly in an object-centered or egocentric reference frame. Figure 2a illustrates reference-object-left and reference-object-right selectivity profiles for two hypothetical neurons, representing idealized object-centered and egocentric reference frames, respectively. In object-centered encoding, spatial selectivity of the neuron only depends on the position relative to the object. Thus, such neuron would show the same pattern of selectivity to different boxes on the object in object-left and -right conditions. Consequently, in egocentric screen coordinates (Fig. 2a, left) this corresponds to a shift of the selectivity profile that matches the object's shift, while the shape of the profile stays the same. On the other hand, if an ideal neuron encodes the cue in an egocentric coordinate system, the selectivity will only depend on the egocentric location of the cue regardless of the object location. By shifting the object on the screen, one would sample different segments of an egocentric selectivity profile. Therefore, when comparing the object-left and object–right selectivity profiles, an

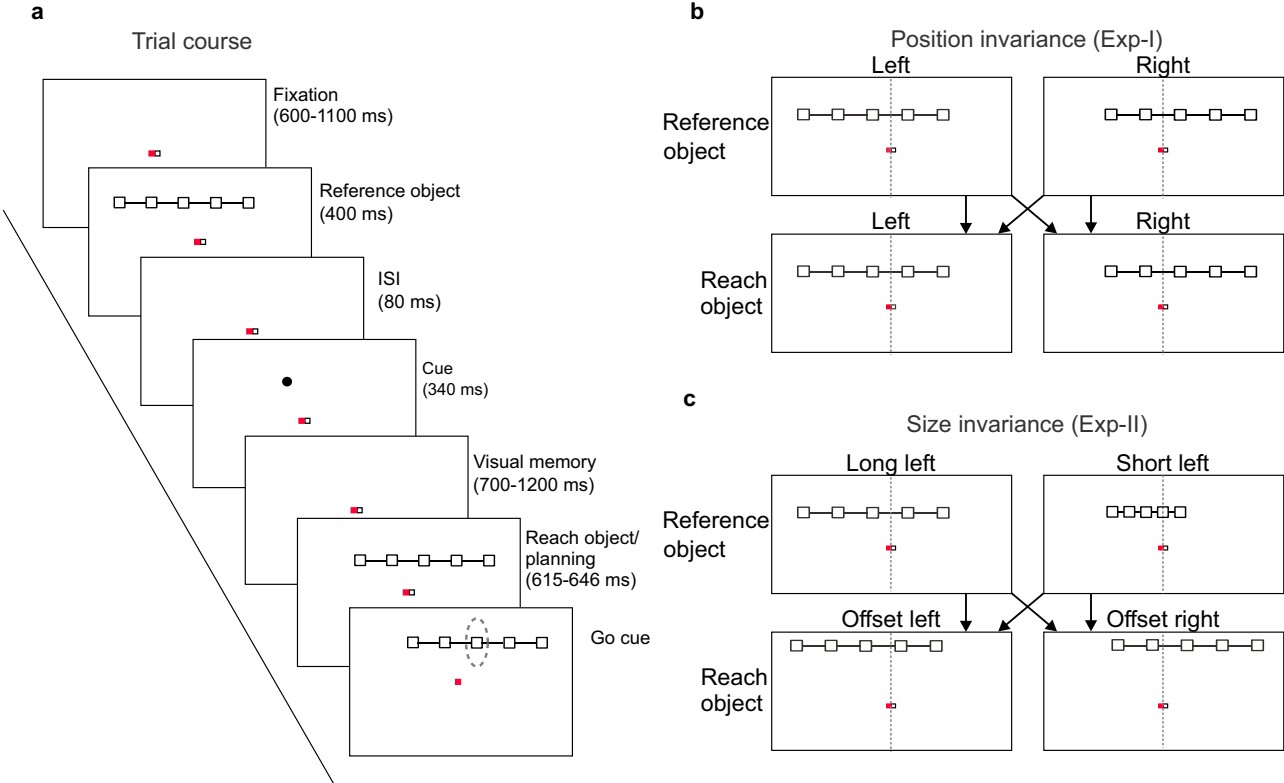

**Fig. 1 | Object-based reach planning task. a** Time course. After acquiring and holding successful ocular fixation of the central red square and touching of the white square during baseline, an array of five interconnected boxes (reference object) is presented randomly with an offset either to the left or right of the screen center. The reference object is followed by a brief cue presentation, located at one of the five box positions (target box), and a visual memory period after which the array of boxes is presented again (reach object), randomly to the left or right of the screen center. During the following delay (movement planning period) the monkey has to maintain ocular fixation and withhold arm movement, thereby keeping body-, hand- and gaze-centered frames of reference aligned with the screen center. Disappearance of the hand fixation stimulus (Go cue) permits the monkey to reach to the memorized target box. The reach goal is defined by the position of the box that was cued on the reference object in object coordinates (dotted ellipse, not shown to animal). Time spans of individual trial periods as written in the labels indicate the range of uniformly distributed random durations. **b** Position invariance (Exp-I). Reference and reach object each are presented at fixed eccentricities

randomly offset to the left or right of the screen center. Left and right offsets are uncorrelated, making the position of the reach object unpredictable and in 50% congruent and 50% incongruent to the reference object. The horizontal offset between left and right object center corresponds to the inter-box distance, so that left and right object location overlap in four of the five boxes. This results in six possible egocentric cue/reach target locations on the screen (Supplementary Fig. 1). In Exp-I, reference and reach objects are always the same size and also otherwise visually identical. **c** Size-invariance (Exp-II). In half of the trials, the reference object had the same length as in Exp-I (condition "long left" in upper row) and in the other trials was half as long ("short left"). Again, reference and reach object each were presented randomly to the left or right of the screen center (see Supplementary Fig. 1 for all possible arrangements). In Exp-II, horizontal offsets of reference and reach object differed in size and an additional small vertical offset was introduced such that reference and reach object were spatially incongruent in all trials (see Methods for details).

egocentric neuron would show the same activity for positions with corresponding egocentric locations (Fig. 2a, right). This logic applies to cue encoding relative to reference object and reach goal encoding relative to the reach object.

Previous studies have reported mixed egocentric encoding in PRR and PMd[13,45,46]. We therefore not only expect diversity in reference frames across neurons, but also combined object-centered and egocentric encoding within individual neurons. A simple case of mixed encoding would be a partial shift of the selectivity profile between object-left and -right conditions, not reflecting the full distance of the horizontal shift. Depending on the shape of the selectivity profile and the weight of either reference frames, more complex mixed-selectivity profiles are conceivable.

**Single units in PRR and PMd encode the locations of visual cues and reach goals in object-centered as well as in egocentric reference frames**

Monkeys K and H performed the task with $72.15\% \pm 1.00\%$ and $87.92\% \pm 1.20\%$ average success rate across sessions (see

Supplementary Fig. 2 for more details). We recorded 100 and 107 single units in PRR and PMd, respectively, from monkey K, and 56 and 57 in PRR and PMd, respectively, from monkey H. From monkey K, 71 (71%) of neurons in PRR and 77 (72%) in PMd were included in the following analyses, from monkey H, 30 (54%) in PRR and 32 (56%) in PMd (see Methods). Since data from the two monkeys yielded corresponding results, throughout, we report the result from combining the two monkeys' data unless mentioned otherwise.

During the visual memory period, selectivity profiles of a subset of cells in both areas were consistent with encoding of the cue location in an object-centered reference frame (Fig. 2b, c, left panels, Example neuron 1). The selectivity profile of this example neuron is independent of the screen-location of the object, hence object-centered. Single units with object-centered selectivity were found in both areas PRR (shown) and PMd of both monkeys.

Selectivity profiles of other units in the same areas and in both animals resembled egocentric encoding of the cue location (Fig. 2b, c, right panels, Example neuron 2). This example unit showed identical neural response strengths when overlapping boxes (i.e., egocentrically

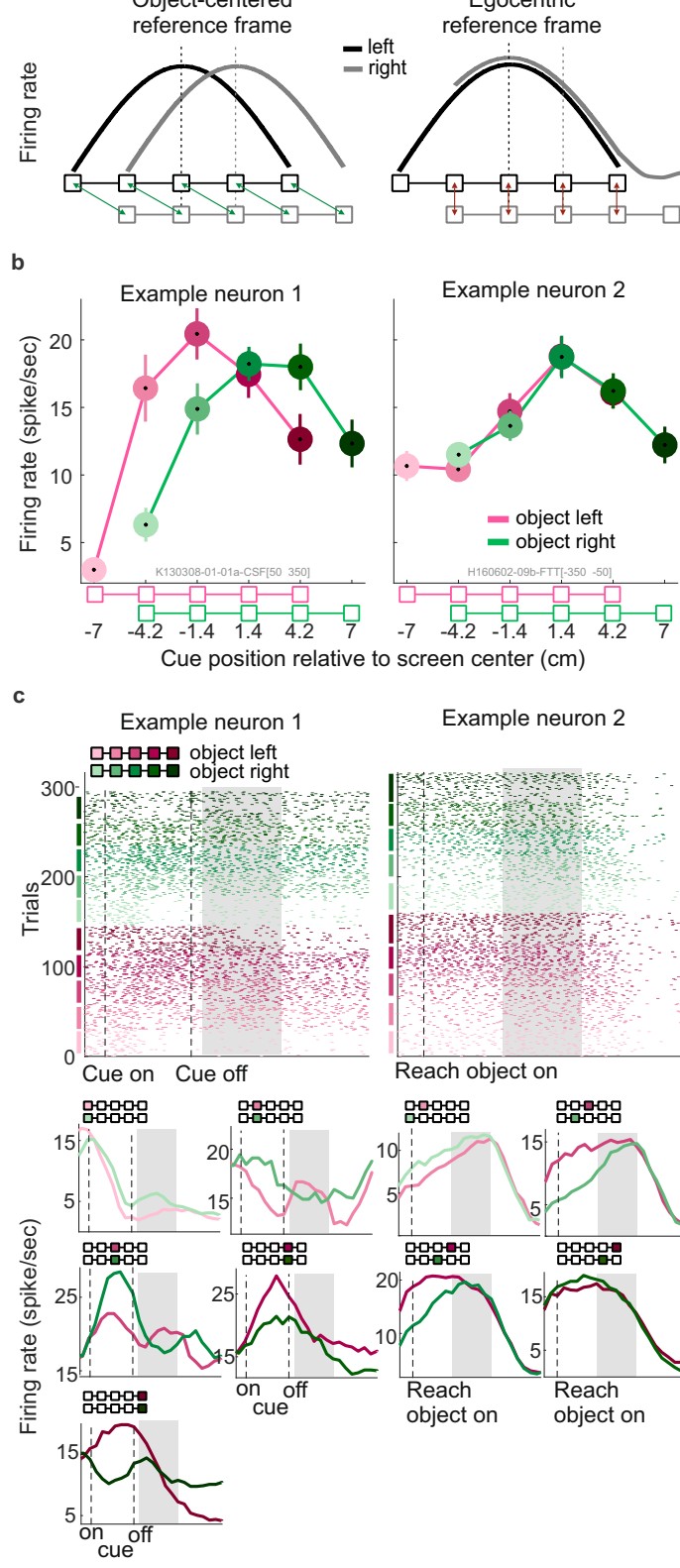

**Fig. 2 | Object-centered and egocentric reference frames. a** Hypothetical object-centered and egocentric single unit responses. The idealized object-centered hypothesis (left column) predicts that a neuron keeps the same selectivity profile for different cue/target locations on the object, independent of the object location on the screen ( = relative to the body). When analyzed as function of screen position, this would result in a shift of the profile together with the object. The egocentric hypothesis (right column) predicts that the selectivity profile is a function of the cue/target location relative to the body and not relative to the object. When comparing object-left and object-right profiles, the neuron shows the same response level for target positions overlapping in screen space, but different activity for non-overlapping boxes. Shifting of the object in this case would mean sampling a different part of the egocentric selectivity profile. **b** Examples of single unit selectivity profiles from Exp-I. Example neuron 1 (monkey K, PRR) shows object-centered encoding (left column) of the cue during early visual memory period, 50–350 ms after cue onset (shaded time window in c). Example neuron 2 (monkey H, PMd) shows egocentric encoding (right column) of the target during planning period, 350-50 ms before movement onset. The curves show mean firing rate across same-condition trials, error bars indicate SEM (number of trials from left to right: example neuron 1, $N_{object-left}$ = [29 28 30 30 29], $N_{object-right}$ = [28 30 30 30 30]; example neuron 2, $N_{object-left}$ = [30 32 34 30 34], $N_{object-right}$ = [31 33 29 30 33]). Different colors correspond to different boxes on the object. **c** Raster plot and peri-stimulus time histogram (PSTH) of the two example units shown in b. Example neuron 1 is aligned relative to the time of cue onset, example neuron 2 relative to the reach object onset. In the raster plots (upper panels), every row is one trial and trials are sorted and grouped according to the on-the-object positions. The PSTHs (lower panels) show firing rates after Gaussian kernel smoothing with 50 ms standard deviation, averaged across trials with identical object and on-the-object cue/target positions. Every panel includes the PSTHs for the two on-the-object cues/targets which according to the object-centered (example neuron 1, left) and egocentric (example neurons 2, right) hypothesis, respectively, correspond to each other. Color conventions as in b. Source data are provided as a Source Data file.

encoding, selectivity patterns of most neurons only partly match the idealized patterns shown in Fig. 2a. This is the case during both the visual memory period and reach planning. Often neurons show more complex patterns with ambiguous or mixed object-centered and egocentric selectivity. Simple examples of ambiguity are linear selectivity profiles, in which case object-left and object-right selectivity profile can result either from a horizontal shift or a vertical (firing rate) offset between both conditions. Examples of mostly linear selectivity profiles are shown in Supplementary Fig. 3a (neuron 1, planning period) and Supplementary Fig. 3b (neuron 2, memory period).

To respect the continuous spectrum reflected in the observed mixed selectivity within and across individual units, we did not attempt to categorize the spatial selectivity of single units as binary object-centered or egocentric classes. Yet, the existence and gradual tendencies for either encoding scheme at the neural population level were quantified to determine whether they differ between brain areas or cognitive states and are informative about underlying computations.

## Predominant reference frame changes from object-centered during visual memory to egocentric during reach planning in PRR and PMd

During the late visual memory period, object-centered encoding of the cue location predominated across the neural populations in both PMd and PRR. We quantified population-level predominance of either encoding scheme over the course of the trial in two ways: first, by computing a position invariance (PI) index, which is based on correlations of spatial selectivity profiles between conditions with different object positions (see Methods); second, with population decoding based on a cross-conditional classifier.

Positive values of the average (across neurons) PI during the memory period indicate predominant object-centered encoding (Fig. 3a). By design, once the location of the reach object is revealed, the egocentric reach goal location is known to the monkeys such that they can use this information for movement planning. After this reach

corresponding cue locations) were cued with the object being located either left or right.

During the movement planning period, we also observed that selectivity profiles of some neurons resembled encoding of the reach goal in the object-centered reference frame and others in an egocentric reference frame (see more example neurons in Supplementary Fig. 3 and Supplementary Table 1). As is not surprising for mixed

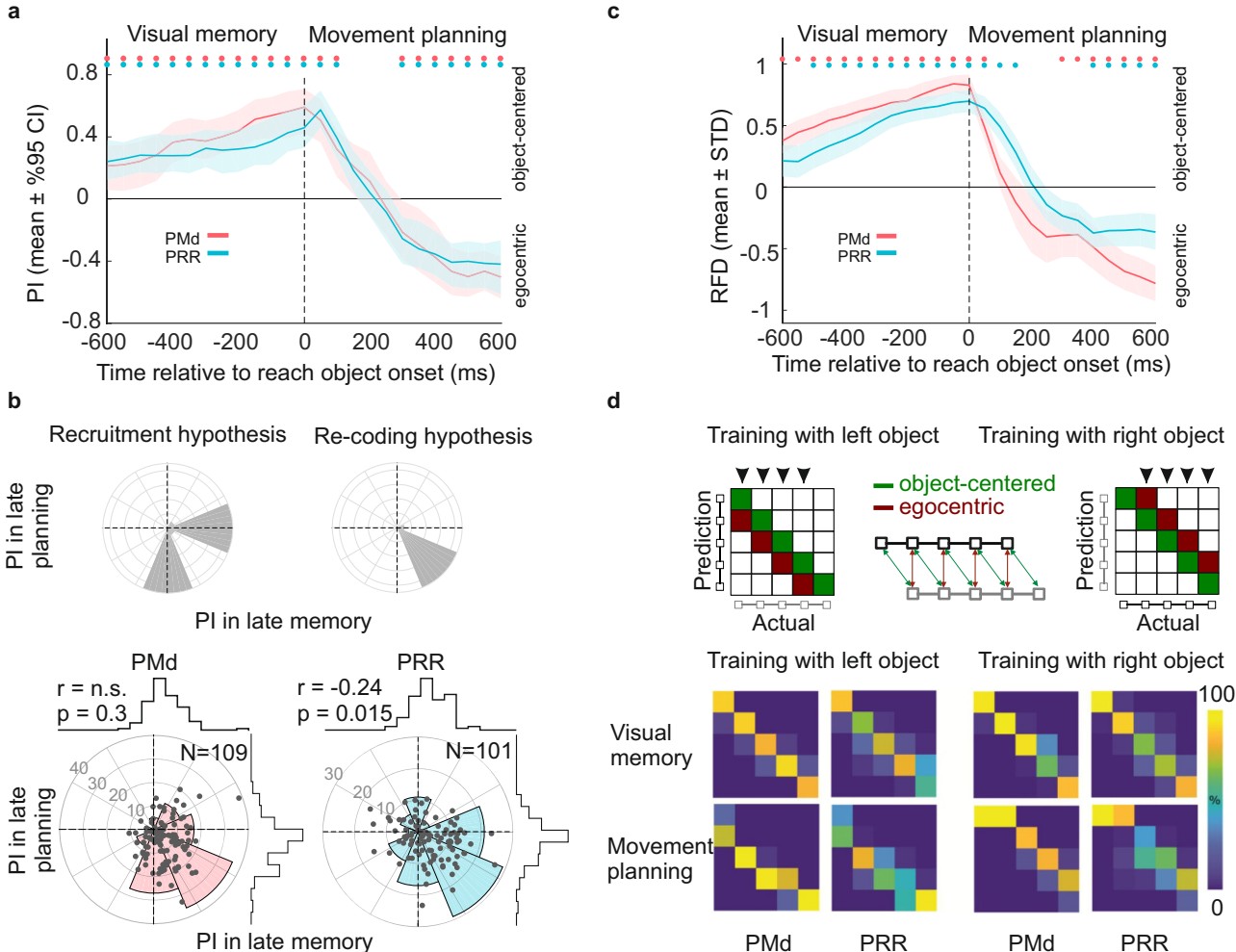

**Fig. 3 | Predominant reference frame in the position invariance task (Exp-I).**
**a** Position Invariance (PI) measure across the population of 101 PRR (blue) and 109 PMd (red) single units (300 ms time bins, sliding by 50 ms, time points on the x-axes indicate center of the time bins). Positive and negative PI values indicate predominant object-centered and egocentric encoding, respectively. In the visual memory period (before reach object onset), object-centered encoding gained higher weight across the population, whereas reach goal locations (after reach object onset) were predominantly encoded in an egocentric reference frame. Dots at the top indicate significant deviation of average PI from zero (t test, p values corrected for multiple comparison across time bins using false discovery rate correction (see "Methods")) **b** Hypothetical (upper panels) and actual (lower panels) reference frame comparison between memory and movement planning period. Top panels show predictions of the recruitment (left) and re-coding (right) hypotheses when PI values of individual neurons are plotted in memory vs movement planning period. Positive and negative values represent predominant object-centered and egocentric encoding, respectively. The recruitment hypothesis predicts that one group of neurons exclusively supports visual memory with object-centered encoding in the memory period, and another group exclusively supports movement planning with egocentric encoding in the planning period. This would result in a bimodal distribution of the angular densities (polar histograms) with peaks at the positive/negative PI axis in the memory/planning period, respectively. The re-coding hypothesis predicts that the same neurons supporting object-centered visual memory during the memory period also support egocentric reach planning during the planning period. This would result in a uni-modal distribution of angular densities with a peak in the lower right quadrant. PI values of individual units (dots in scatter plot) in late memory (300 ms before reach object onset) vs late reach planning period (300–600 ms after reach object onset) are shown in the bottom panels. Horizontal and vertical histograms show marginal distributions of the PI values on the two axes. Angular densities (polar histograms) across the population of neurons show a peak in the lower right quadrant and do not indicate deviations from a unimodal distribution (Hartigan's dip test: PRR $p = 0.84$; PMd $p = 0.85$, one-sided test). r- and p values in the plot indicate Pearson's correlation coefficient and its significance level. **c** Relative difference of reference frame decoding (RFD). Positive and negative RFD values indicate predominant classification of object-centered and egocentric positions, respectively. Dots at the top indicate significant deviation from zero (randomization test for RFD > 0, PRR $p < 0.001$; PMd p < 0.001, two-sided test, randomization test, see "Methods" section; p values corrected for multiple comparison across time bins using false discovery rate correction). **d** Hypothetical (upper panels) and actual (lower panels) confusion matrices in cross-conditional decoding. A 5-way classifier was trained to classify the object-centered position of the target box in trials when the object was on one side of the screen. In the cross-conditional decoding approach, the classifier was then tested on neural data when the object was on the other side of the screen. Good classifier performance indicates object-centered encoding. In the hypothetical confusion matrix (top panel), correct classifications fall on the main diagonal (green). An egocentric encoding, instead, would introduce a systematic misclassification of the data, resulting in confusion matrices filled along the diagonals that are one off the main diagonal to the left or right (red), depending on which side is used for training the classifier. Arrows on top of the matrix mark the task conditions that are included in the calculation of the RFD values, as they have a distinct egocentric and object-centered representation (see Methods). Bottom panel shows actual confusion matrices. During the late memory period, object-centered positions were classified accurately. During late movement planning, a shift off the diagonal indicates classification of egocentric positions in both brain areas. Source data are provided as a Source Data file.

object onset, the average PI shifted within 300–500 ms, from predominant object-centered to predominant egocentric spatial selectivity in both brain areas (Fig. 3a, Supplementary Fig. 4 shows monkeys separately).

The single unit analysis across the population of neurons allows to test if the distribution of PI values shifts due to two complementary neuronal subpopulations, each of which selectively contributes to spatial encoding only in the memory or in the planning period, respectively. According to this "recruitment" hypothesis, one group of neurons exclusively supports visual memory with object-centered encoding in the memory period, the other group exclusively supports movement planning with egocentric encoding in the planning period (Fig. 3b, top row, left column). As an alternative hypothesis, the population-level shift in PI may be the consequence of a consistent shift in reference frame of each individual neuron. According to this "re-coding" hypothesis (Fig. 3b, top row, right column), the same neurons supporting object-centered visual memory during the memory period (positive PI) also support egocentric reach planning during the planning period (negative PI).

To test these two hypotheses, we compared PI in visual memory versus movement planning and quantified across the population of neurons if a strong encoding of one reference frame during visual memory would be associated with a strong encoding for the same, the other, or no reference frame during movement planning. We plotted the PI of each neuron in memory vs planning period on orthogonal axes (Fig. 3b, bottom row), and measured the angular deviation θ from the x + axis, between −180 and 180 degrees, corresponding to the arctan of the ratio PI_planning/PI_memory. Neurons showing a reference frame encoding selectively in only one delay period fall onto one of the main axes; neurons with strongly (anti-)correlated reference frame between visual memory and movement planning fall on the diagonals. Predominant subpopulations of either type would result in an inhomogeneous distribution of the angles. Distributions were estimated by binning the data into 45° bins, and subjected to Hartigan's dip test for multimodality[49]. We compared responses 300 ms preceding (late memory) and 300–600 ms after (late planning period) reach object onset, when the neural activity is comparably stable after the transient change in stimulus. The recruitment hypothesis predicts a bimodal distribution with the two lobes pointing to the right-horizontal axis and the bottom-vertical axis, respectively. The re-coding hypothesis predicts a unimodal distribution pointing to the lower right quadrant.

The PI values cluster in the lower right quadrant (Fig. 3b, mean ± SEM PRR along memory period axis 0.37 ± 0.07, movement planning axis −0.42 ± 0.08; PMd along memory period axis 0.51 ± 0.07, movement planning axis −0.51 ± 0.08) and the angular density distribution is unimodal, as expected for the re-coding hypothesis. There is no indication of a clustering along the main axes and a corresponding bimodal angular density distribution, as would be expected for the recruitment hypothesis (Hartigan's dip test[49], PRR $p = 0.84$; PMd $p = 0.85$). This re-coding pattern suggests that in PRR and PMd, the population of neurons contributing to either predominant reference frame in the two different periods mostly overlap. There is a weak but statistically significant negative correlation between PI of neurons in late memory period and movement planning period in PRR but not PMd (Pearson's correlation coefficient; PRR $r = −0.24$, $p = 0.015$; PMd $p = 0.3$). This suggests that at least in PRR, on average, units with stronger object-centered preference in late memory tend to be more strongly egocentric in the later planning period.

Second, we used neural population decoding to quantify the predominance of either reference frame. For this, we applied cross-conditional classification, and characterized generalization errors. Classifiers were trained to distinguish the five positions on the object using only trials where the object was on one side and tested its performance on trials where the object was on the other side. Predominant egocentric or object-centered neural reference frames,

respectively, then predict characteristic classification patterns. High classification accuracy suggests an object-centered population encoding since each position on the object is decoded correctly irrespective of the object location relative to the body. Egocentric encoding instead would introduce systematic misclassification, namely a shift by one position off the main diagonal in the confusion matrix (Fig. 3d, top row).

From monkey K, 83–84 (83–84%) of neurons in PRR and 99-101 (93–94%) in PMd were included in the decoding analysis based on the number of available trials per condition (see Methods), from monkey H, 39–41 (70–73%) in PRR and 43-45 (75–79%) in PMd. There is a small variability in the number of neurons between conditions because the exact number of available trials depends on the condition that was used for training the decoder. When tested on the same data as used for training (iso-conditional), the classifiers showed high cross-validated classification accuracy of 95% for PMd and 80% for PRR on average over time bins and trial subsamples. To test for the predominant reference frame, we computed the difference between the percentage of test trials classified in accordance with the object-centered hypothesis and the percentage classified in accordance with the egocentric hypothesis during the cross-conditional decoding (reference frame difference RFD; see Methods). The RFD (Fig. 3c) and confusion matrices (Fig. 3d, bottom row) show that during the visual memory period the cue position is predominantly classified in accordance with an object-centered reference frame (randomization test for RFD > 0, PRR $p < 0.001$; PMd $p < 0.001$; randomization test, see Methods section). After showing the reach object there is a transition to classifying positions predominantly in an egocentric reference frame in the late movement planning phase (randomization test for RFD < 0, PRR $p = 0.004$; PMd $p < 0.001$; randomization test).

## Single units scale the width of their selectivity profile with object size in PRR and PMd

Our results from Exp-I suggest a predominant object-centered encoding of the cue location during the memory period across the neuronal population. In Exp-II, we asked if during this time, the fronto-parietal network can encode on-the-object locations in an object-centered manner also for variable object size. Real-life objects might exist in different-size variants, like hammers for different purposes. Hence, ideal object-centered encoding of on-the-object locations predicts that selectivity profiles of neurons should be characterized only by the relative positioning of the cue and the object, irrespective of object location (origin) and object size (scale), i.e., by position-invariant and size-invariant object-centered selectivity. Since we characterize neural selectivity along the horizontal spatial dimension in this study, we also scaled the objects only along this one horizontal dimension, i.e., in their length. In Exp-II, the monkeys saw a long or a short reference object, in randomly interleaved trials, relative to which they had to memorize the cue location (Fig. 1c). In order to preserve object-centered position information, one would expect a compression of the selectivity profile (in the egocentric screen space) for object-short compared to object-long trials (Fig. 4a, left). In contrast, egocentric encoding would result in sampling of a reduced range of the selectivity profile in object-short trials (Fig. 4a, right).

We tested for size invariance only during the visual memory period, not during movement planning, since Exp-I had shown predominant object-centered encoding during the memory period only. Consequently, in Exp-II, only variation of the size of the reference object was relevant. Therefore, reference objects were either short or long from trial to trial, while reach objects were always long to make it easier for the animals to hit the target boxes correctly (Fig. 1c).

For Exp-II, we recorded 67 and 53 single units in PRR and PMd, respectively, from monkey K; 36 and 45 single units in PRR and PMd from monkey H. We identified 51 (76%) and 46 (87%) active units in PRR and PMd from monkey K, 19 (53%) and 20 (44%) from monkey H, which

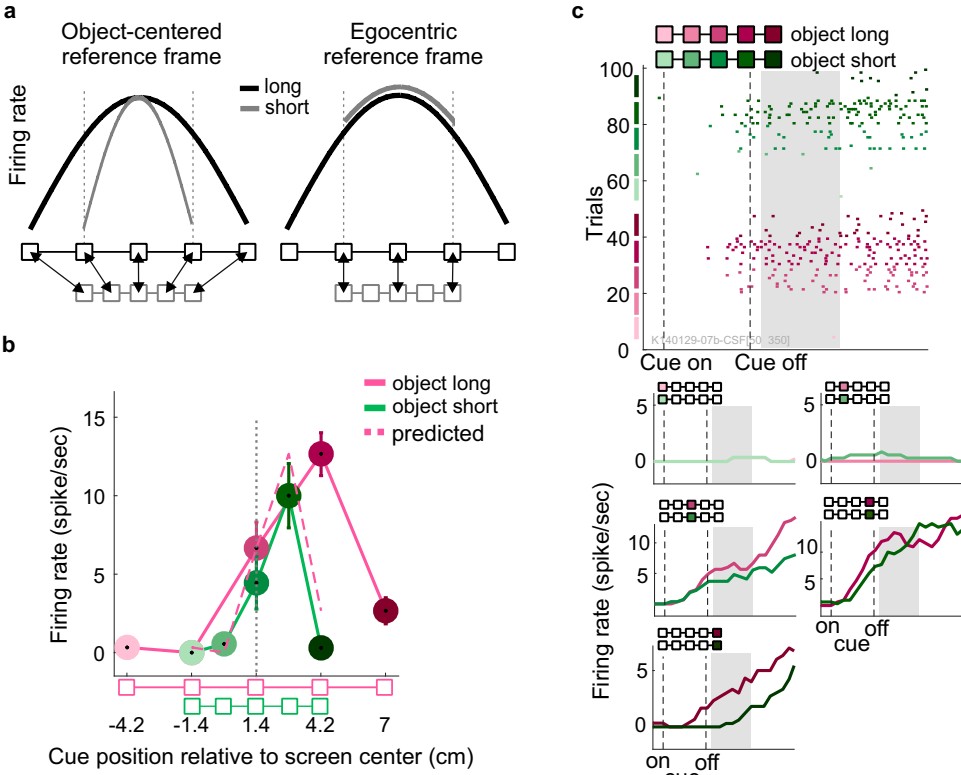

**Fig. 4 | Size-invariant object-centered encoding (Exp-II). a** Hypothetical size-dependent and size-invariant single unit responses. Selectivity profiles of idealized object-centered (size-invariant, left) and egocentric (size-dependent, right) neuronal selectivity profiles. The size-invariance hypothesis predicts that the selectivity profile scales with the object size, such that the profile to different locations on the object is the same irrespective of object size. The egocentric hypothesis predicts the same response level for positions with identical egocentric locations, thereby sampling less horizontal extend when objects are small. **b, c** Example unit from Exp-II. Activity of an example unit when short and long reference objects were

presented at the right side of the screen center. Object-centered size-invariant encoding is identified by a horizontally scaled selectivity profile. The dashed curve represents the ideal object-centered scaling of the selectivity profile for short-object trials, predicted from the selectivity profile observed in long-object trials, assuming that the width of the activity profile was scaled with the size of the object and relative to center of the object. Other conventions as in Fig. 2b, c. Number of samples for calculating SEM for left to the right boxes: $N_{\text{long object}} = [10\ 10\ 10\ 10\ 10]$, $N_{\text{short object}} = [9\ 12\ 9\ 10\ 11]$. Source data are provided as a Source Data file.

were included in the following analysis. Monkeys K and H performed the task with 63.94% ± 0.88% and 81.70% ± 2.50% average success rate (Supplementary Fig. 5 for more details).

Neurons in PRR and in PMd of both monkeys revealed examples of size-invariant object-centered encoding during visual memory. Figure 4b, c shows an example unit where the object-short response profile fits the prediction of the object-centered hypothesis in the sense that it represents a duplicate of the long-object selectivity profile compressed by the factor that short and long object differ in length. The example neuron showed the same selectivity to long and short reference objects, respectively, when they were presented on the left of the screen center (not shown). In other words, only the relative on-object position determined the response, not the location of the object on the screen nor the size of the object. The actually measured profile in this example precisely matched the prediction (the dashed curve) except for a small reduction in response gain by about 10%. If normalizing to the maximal response, result and prediction match exactly (not shown).

As in Exp-I, also in Exp-II we observed units with more complex selectivity profiles which either do not or only partly fit the object-centered or the egocentric hypothesis (Supplementary Fig. 6).

In order to assess the contribution of position- and size-invariant encoding at the neural population level, we again computed the correlation measure and the decoding measure. For the correlation measure, we computed PSI for each neuron's selectivity profiles (see Methods). The distribution of PSI across units (Fig. 5b) shows that different units scale their selectivity profile to different extent, evident

from the range of PSI values across the population. However, on average, size-invariance predominated (signed-rank test, PRR z = 5.80, p < 0.001; PMd z = 6.08, p < 0.001), consistently across areas.

With the cross-conditional classification approach, we tested for generalization of the decoder between the long and short reference object conditions (Fig. 5a, see "Methods"). The cross-validated accuracy of the classifiers on the training data was high during the visual memory period, 88% for PMd and 80% for PRR on average over time and repetitions. The RFD in Fig. 5c shows that in both areas the cue position predominantly can be decoded invariant to the size of the object during the late visual memory period (randomization test for RFD > 0, PRR p < 0.001; PMd p < 0.001).

When testing each monkey individually, both showed significant results supporting size invariance in Exp II. In monkey K, these findings were highly significant across all measures. In monkey H, for which only a lower number of neurons could be recorded in Exp II, findings show partially significant shifts (PSI), else the same trends (RFD) as in monkey K in favor of size invariance. Additionally, both animals show highly significant support for object-centered (size invariant) encoding when decomposing the neural population dynamics with a demixed principle component analysis (dPCA). Supplementary Note 1 and Supplementary Fig. 8 and 9 show the results of Exp II for the individual monkeys.

## Discussion

Reaching and grasping are major ways in which primates interact with objects in their environment. Object-relative position information is

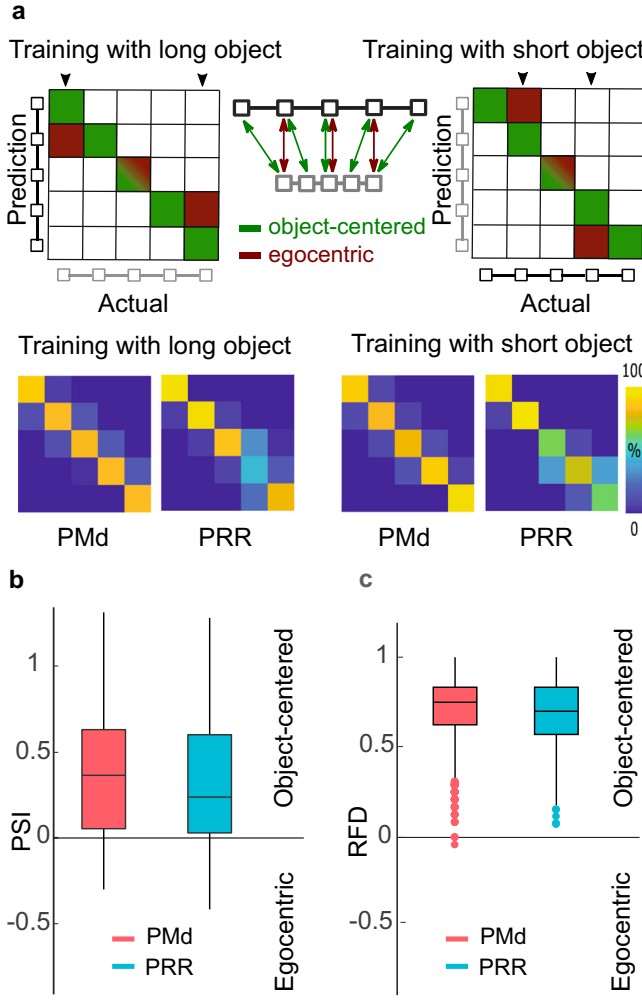

**Fig. 5 | Predominant object-centered encoding across neuronal populations in the size invariant task. a** Hypothetical and actual confusion matrices in cross-conditional decoding. Equivalent to Fig. 3d, a 5-way classifier was trained in trials with one object size and then used to classify the object-centered position of the target box when the object had the other size. In the hypothetical confusion matrix (upper panels), these trials fall on the main diagonal (green). An egocentric encoding, instead, would introduce a systematic misclassification of the data, resulting in diagonals that are compressed compared to the main diagonal, vertically or horizontally (red), depending on which object size is used for training the classifier. Arrows on top of the matrix mark the task conditions that are included in the calculation of the RFD values, as they have a distinct egocentric and object-centered representation (see Methods). The actual confusion matrices (bottom rows) show object-centered (size invariant) encoding. **b** Across the population of PMd (red) and PRR (blue) neurons, Position and Size Invariance (PSI, see Method) values were mostly positive, indicating predominant size-invariant neural selectivity in the last 300 ms of visual memory period (two-sided signed-rank test, PRR $p < 0.001$, PMd p < 0.001; number of samples $N_{PRR} = 70$, $N_{PMd} = 66$). **c** Equivalently, positive RFD values indicate predominant classification of object-centered rather than egocentric positions, respectively (randomization test for RFD > 0, PRR $p < 0.001$; PMd $p < 0.001$, two-sided test; $N = 1200$ samples for each area). The box plots show median and the 75th (top) and 25th (bottom) percentiles, as well as the data range (whiskers) without putative outliers (dots; more distant from 25/75 percentiles than 1.5 times the respective interquartile range). Source data are provided as a Source Data file.

we showed, first, when localizing spatial targets to direct a reach towards different on-the-object sites, object-centered encoding is part of the mixed ego- and allocentric neural selectivity found in the frontoparietal reach planning network. Second, in both, PRR and PMd the predominant frame of reference shifts according to cognitive demands in different epochs of the behavioral task. Within trials, during visual memory, when the cue-on-object location was the most parsimonious way of maintaining the relevant spatial information, object-centered encoding predominated. Later, during movement planning in the same trial, when the actual reach goal could be determined from the integration of the memorized cue-on-object location and the now visible and stationary reach object location, egocentric encoding predominated. Third, by providing evidence for size-invariant spatial selectivity, we demonstrate that allocentric encoding not only means that the origin of the reference frame is centered on the object, but also that the spatial scale scales with object-size. Such size-invariant representation of object-related information so far was associated with object-recognition tasks and mainly ventral stream processing[51–53], but not with dorsal stream movement preparation.

We see that ego- or allocentric reference frames can dominate within individual areas of the frontoparietal reach network during preparation of skeletomotor movements. The allocentric frame dominated while the animals had to memorize an on-the-object position to generalize this geometric information to the time when the final object location on the screen would become available. The egocentric frame dominated as soon as animals could plan an according reach movement, reminiscent of human imaging data[23]. Note that due to the invariant position and starting pose of the animals in our experiment, we cannot fully rule out that the observed egocentric encoding is (partially) world-centered, hence non-egocentric. Yet, since previous experiments reported predominant egocentric encoding during reach planning in PRR and PMd[12–14], we refer to the here observed non-objected-centered encoding as egocentric.

While static mixed egocentric reference frames in neurons have been described before[18,38,45,54], our findings further provide evidence that the predominant reference frame is not a fixed property in frontoparietal association cortices, not even at the level of individual neurons. Shifts between different egocentric frames have been observed before in parietal areas in the context of arm movement preparation in human imaging[55] and also monkey neurophysiology[39]. Yet, shifts from allo- to egocentric encoding during movement planning so far have been suggested from human imaging data only, and then in different parts of the brain. While two studies[22,23] reported strongest allocentric preference in temporal cortices, they attribute mixed ego- and allocentric encoding for computing allo-to-ego conversions to human dorsal premotor cortex (PMd), pre-supplementary motor area (pre-SMA) and precuneus, but not to medial intraparietal sulcus (mIPS), the superior parietal lobe (SPL) or the superior parieto-occipital (SPOC), which were predominantly egocentric[9,22,23]. At the individual neuron level, in a perceptual judgment task, individual VIP cells were shown to encode visual pattern motion direction in head-centered (egocentric) or world-centered (allocentric) reference frame, depending on the trial-to-trial instructions[18]. During saccade preparation, neurons in frontal eye fields (FEF) showed gaze-planning signals with an impact of allocentric landmark information that varied during different stages of action preparation. Here we showed that flexible shifting between allo- and ego-centric frames of reference at the neuron level is not limited to context-dependent perceptual judgment based on multi-sensory integration or saccade preparation in landmark tasks in the frontal lobe, but also applies to goal-directed movement planning signals in posterior parietal cortex.

Our results also demonstrate that such allo-to-ego transition occurs without externally driven explicit task instructions, but as consequence of a within-task dynamic switch in spatial cognitive

important in guiding these movements. Such allocentric alongside egocentric frames of reference provide a stable representation of space which is robust to environmental dynamics and noise[50], for example, when the reach goal is located on a moving object which is temporarily occluded and reappears in unpredictable location. Here

demand. With the same task instruction and within the same trial, the cognitive strategy that controls movement planning using sensory information determines the predominant reference frame in PRR and PMd. The diversity in encoding provides high flexibility for the fronto-parietal network for conveying spatial information to recipient areas in either of the two frames of reference. The observed dynamics reflect task-specific computations to achieve spatial transformation[10], thereby not just giving variable weight to different egocentric sensory modalities, but also including allocentric computations if demanded by the task. In our decoding analyses, PMd reflected the allo-to-ego transition earlier than PRR (Fig. 3c), while correlation-based measures (Fig. 3a), did not indicate a latency difference between both areas. This means, if there is a latency difference at all, then PMd leads the shift in reference frame. Such finding would support earlier conclusions that frontal areas lead parietal areas in determining motor goals when tasks require a spatial remapping[27], and the more general notion that frontal lobe exerts cognitive control over posterior regions of the brain.

For successful manual interactions with an object, a geometrical representation of the object, including features such as object size and distance between on-the-object sites, is necessary for action planning. Additional to pre-shaping the hand to fit the geometry of a physical object before grasping[56], a proper object-relative placement of the hand is needed, e.g., to pick up a hammer at its handle instead of its head. This is also true for visual 2D objects, e.g., when dragging digital windows of different sizes or positions on a touchscreen by touching them with the finger at the top bar. Correct object-relative hand positioning based on object geometry is relevant for successful completion of the object-associated action, and hence might differ from allocentric localization of reach goals relative to a visual landmark with which one is not going to have direct physical interaction and could even be outside the peripersonal space. This difference might explain why the frontoparietal network showed higher BOLD activity for allocentric compared to egocentric movements when human subjects had to estimate reach goal locations based on geometrical features (like relative distance and direction of a remote pair of dots)[24]. When instead a target had to be localized relative to a visual landmark (left or right of the landmark), spatial processing along the dorsal stream was predominantly egocentric[22]. Our task also requires interaction with the two dimensional visual object on the screen, which might explain why we see object-centered encoding.

The relevance of allocentric, especially object-centered encoding in skeletomotor compared to oculomotor tasks was unclear so far, since detailed monkey neurophysiology data existed only for saccade tasks. While it has been shown that, the frontal eye fields (FEF) have an active role in integrating the position of the saccade target relative to gaze and a task-irrelevant landmark during saccade planning[57], neurons in the lateral intraparietal area (LIP) did not show predominant object-centered encoding during an object-relative saccade task, which was otherwise similar to our task[58]. The importance of object geometry for manual interaction with objects may explain why we found allocentric encoding in our reach task.

On the other hand, saccade studies in monkey parietal area 7a suggest that individual neurons can encode target location not only in gaze-centered (egocentric) reference frame[59], but also show left-right selectivity for spatial information relative to task-relevant objects[60,61]. Similarly, subsets of neurons in the supplementary eye fields (SEF) of the frontal cortex can be selective for the left or right end of an object or a pair of dots towards which a saccade is directed[62–66]. Yet, for binary left-right selectivity, it can be difficult to properly distinguish categorical, rule-like encoding from an object-relative position code. Given that neurons, especially in the frontal areas, are known to represent categorical abstract rules[67,68], it was speculated if object-relative left-right neural selectivity could result from top-down rule signals in previous object-centered saccade experiments[8]. In our task, we sampled space-continuous selectivity profiles with five on-the-object positions. Rule encoding would predict identical selectivity profiles for object left/right/long/short conditions, which was not the case for most of the units we recorded. Instead, we observed a spectrum of ego- and object-centered encoding, including mixed reference frames and partial scaling, as typical for the computation of coordinate transformations in neural networks[10,11,69–71], which cannot be explained merely by rule encoding. We therefore interpret our observed neural selectivity profiles as indication of spatial encoding supporting the transformation between ego- and allocentric representations of space in the context of planning skeletomotor movements, rather than categorical rule encoding.

Remarkably, object-centered encoding was accompanied with size-invariant spatial encoding of different on-the-object sites, at the single neuron and population levels. To our knowledge, before this study, there was no electrophysiology evidence showing that the fronto-parietal network utilizes size-invariant positional code for movement planning. Size-invariance is a coding property discussed in the context of object recognition and associated with ventral stream processing[51–53]. Yet, the relative position of object features is not just relevant for recognition, but also for interaction with the object, e.g., to pick up different-sized hammers always at the end of their handle. Since we presented reference array and cue not simultaneously (in one monkey), we consider it unlikely that the animal memorized the position information by means of a visual pattern and that the size-invariant selectivity observed here "echoes" ventral stream pattern encoding.

Previous studies showed generalized quantity encoding in neurons in the depth of the intraparietal sulcus of monkeys[72] and in parietal BOLD signals in humans[73], i.e., selectivity for quantity irrespective if quantity was presented via the size of an object or the number of visual items. Due to the discrete nature of the on-the-object cue positions, it is possible that the animals memorized them using an object-relative approach (left-most, left, middle, …) or an abstracted positional code (1st, 2nd, 3rd,… position on the object) which is numerical but still object-related. Translating back such abstracted numerical information into spatial motor goal information would be a highly useful capacity, e.g., for foraging in discretized environments ("5th tree from the left").

PRR and PMd are along the course of the dorsal visual processing pathway and we showed that they express allocentric and size-invariant encoding with a short latency of a few hundred milliseconds after presentation of an object-relative spatial cue. Allocentric and size invariant abstract representations are two important properties which according to the classical two visual pathways model[74–76] are associated with the areas along the ventral stream. The model suggests that in memory-guided actions, the dorsal stream could access the ventral stream neural codes through interaction with different areas in the ventral stream. However, it does not predict the latencies of the interaction and how quickly the codes could be accessible in the dorsal stream. If the model accounts for our results, as we had predicted in our earlier behavioral study[77], our findings strongly suggests that the interaction across the two pathways are almost immediate and does not take the long latencies which had been previously suggested by behavioral experiments (see[78] for review). Functional and direct and indirect anatomical connections between frontal and parietal cortex with areas of the ventral stream exist[79–81]. Our data suggest that in order to plan movements which take higher levels of computations and cognitive load for interacting with objects according to geometrical considerations, the two processing streams need to be functionally tightly connected and work together as a network to support the dorsal stream for flexible action planning to interact with non-stationary dynamic environments.

## Methods
### Object-directed memory-guided reaching task
The monkeys were seated in a primate chair in a dimly lit room in front of a fronto-parallel touchscreen. With the help of head-fixation and

trained gaze fixation (see below), the monkeys' mid-sagittal plane and all egocentric references were aligned to the screen center. This means, in the context of our experiments, we did not dissociate different egocentric frames of reference (trunk-, head-, hand- or gaze-centered). Also, these egocentric frames of reference are congruent with a world-centered reference frame, since the animal's position and orientation in the room did not change. Visual stimuli were presented on an LCD screen (19" ViewSonic VX922; onset latencies corrected; background intensity of 0.16 cd/m2) mounted behind the touchscreen (IntelliTouch, ELO Systems, CA, USA). The distance between the monkeys' eyes and the screen was 39-45 cm. Through-out the text, conversions from centimeter to degree are based on a 40 cm distance.

The temporal structure of the task was identical in Exp-I and II (Fig. 1a). The monkey initiated a trial by acquiring central gaze fixation (224 Hz CCD camera, ET-49B, Thomas Recording) and hand fixation on the touchscreen. The gaze fixation stimulus was a filled red square of 0.5 cm (0.72°) side length and 7 cd/m² intensity, and the hand fixation stimulus was a filled white square of 0.5 cm (0.72°) side length and 13 cd/m² intensity. Gaze and hand fixation was enforced within 2–3 cm (2.86–2.89°) around each of the two immediately adjacent fixation points. In case of unsuccessful fixation, the trial was aborted and repeated at a random later time during the experiment.

Valid gaze and hand fixation for a random fixation period of 600–1100 ms was followed by a 400 ms presentation of an array of five boxes, horizontally arranged and connected with a line (reference object; details see below). The boxes indicated possible positions of the upcoming spatial cue. The cue consisted of a small dot of 0.27 cm (0.39°) diameter presented at the position of one of the five reference object boxes. To balance task conditions, cue positions were selected pseudo-randomly with increasing probability of so-far under-represented conditions, and such that the difference in number of correct repetitions between different positions did not exceed three trials.

After an inter-stimulus interval (ISI) of 80 ms following the offset of the reference object, a spatial cue was presented for 340 ms. For monkey H, the ISI was removed and the reference object remained visible during the 420 (= 80 + 340) ms of cue presentation time. The cue presentation was followed by a first variable memory period of 700-1200 ms for monkey K and 900-1200 ms for monkey H, during which only the fixation stimuli were shown (visual memory). After this delay, a reach object of the same type as the reference object was presented. The monkey was instructed to later touch the box on the reach object which corresponded to the box that was cued on the reference object, e.g., for a cue seen at the left-most box of the reference object, the monkey should reach towards the left-most box of the reach object (target), irrespective of the absolute position of the reach object on the screen.

The onset of the reach object was followed by a second delay period of 615–646 ms during which the reach object was visible but the monkey was required to maintain gaze fixation and to withhold the movement (movement planning). Continued visibility of the reach object allowed the animals mentally maintaining the reach goal location either in an object-centered or egocentric reference frame. After the second delay period, the hand fixation stimulus disappeared. This served as the GO cue to reach to and touch the cued on-the-object target location within 1000 ms while holding central ocular fixation. Reach endpoints had to be within an elliptical area (horizontal semi-minor axis 1.2 cm (1.72°); vertical semi-major axis 4 cm (5.71°)) around the target box. After holding the target for 220 ms (monkey H; 300 ms monkey K) the trial counted as successful and the monkey received a visual (a small, light gray dot of 0.27 cm (0.39°) diameter on the target box of the reach object), an acoustic (a high-pitched tone) feedback, and a drop of juice as reward.

The monkey's mid-sagittal plane, gaze and hand fixation points aligned to the center of the screen. The reference object was randomly presented at one of two egocentric locations, left or right of the screen center, with equal eccentricity (Fig. 1b). The reach object was also presented either left or right, but the possible locations differed across experiments I and II (see below). Here and throughout the text, "location" of the object refers to the position of the center of mass of the object on the screen, i.e., the center point of the central box on the object. Reference object, reach object and cue had a low intensity gray tone (2 cd/m²).

## Exp-1 position invariance

In Exp-I, reference and reach objects were always identical in terms of visual appearance. The individual boxes of the reference and reach objects were 0.35 cm (0.50°) squares with 2.8 cm (4.00°) center-to-center distance. In terms of location, reference and reach object either matched (position-congruent trials) or differed (position-incongruent). Left and right screen locations were (x,y) = (± 1.4, 2.7) cm = (± 2.0, 3.8) ° relative to screen center. This means, objects were vertically elevated above the eye and hand fixation position to prevent visual interference with fixation stimuli and obstruction by the animal's arm. Horizontally, in position-incongruent trials, the locations of the reference and reach object boxes were set off by one box distance, such that four out of five box positions overlapped between reference (potential cue locations) and reach object (potential reach targets). For example, boxes 2 to 5 (counting from left to right) of a left-side object had identical egocentric locations to boxes 1 to 4 of a right-side object (Fig. 1b). Therefore, while the cue (target) could take five different positions relative to the reference (decision) object, i.e., five different object-centered positions, they covered in total six different potential egocentric locations on the screen. The 20 different combinations of cue, reference and reach object positions (5×2×2) were presented pseudo-randomly (algorithm as above). By the nature of the behavioral task, the cue and the reach goal always had the same object-centered position. In position-congruent trials, the cue and the reach goal additionally had the same egocentric locations. In position-incongruent trials, the cue and the reach goal differed in egocentric position and the reach goal needed to be determined based on the object-based location of the target box. Since the congruency of the trials was unpredictable, the monkey was encouraged to follow object-centered encoding of the cue in all trials for successful task performance. The monkeys could only determine the egocentric reach goal location upon occurrence of the reach object.

## Exp-2-size invariance

In Exp-II, reference and reach objects always differed in terms of location (position-incongruent trials only) and could additionally differ in size. In long-object trials (size-congruent, Fig. 1c, top left), the reference object was identical to Exp-I. In short-object trials (size-incongruent, Fig. 1c, top right), the boxes of the reference object were only 1.4 cm (2.00°) apart, i.e. half the spacing of the long object. The reach object in both long and short trials was identical to the long reference object. Both long and short reference objects were presented at the same left and right locations as in Exp-I. Unlike Exp-I, the reach object in Exp-II was always position-incongruent to the reference object. The reach object always had a horizontal and vertical position offset and was located at (x,y) = (± 2.3, 4.4) cm = (± 3.4, 6.3)° (Fig. 1c, bottom). This offset was introduced to maximally encourage object-based encoding of the cue location during the visual memory period, since the purpose of Exp-II was to specifically test for scale invariance during object centered encoding. Again, by nature of the task, the cue and the reach goal always had the same object-centered position. Long- and short-object trials were presented in alternating blocks. In long-object blocks, as in Exp-I, all 20 different combinations of cue, reference and reach object positions (5×2×2) were pseudo-randomly presented. In short-object blocks, reference object-left and -right conditions were blocked for ease of performance. In each sub-block,

the 5×2 combinations of cue and reach object locations where randomly interleaved such that horizontal (in-)congruency was always unpredictable and hence the reach goal unknown prior to onset of the reach object.

## Behavioral success rate

was calculated as percentage of correctly performed reaches relative to the number of trials that the monkey initiated and in which it attempted to perform the task by initiating a reach movement. Successfully initiated but unsuccessfully completed trials include cases of belated start of the reach, touching the wrong box on the object, touching out of the predefined tolerance window around the target box, touching the target box but not holding the position long enough, or breaking eye fixation.

## Single unit selection

In both experiments, only "active" units were included in the reference frame analysis and were defined as units which on average (across all trials, regardless of task condition) fired at least 5 spikes within a time interval extending from 100 ms before until 1500 ms after cue onset in correctly performed trials. Average firing rate across all selected units in Exp-I for PRR and PMd was 15.63 and 15.02, respectively, in monkey K; 12.03 and 10.66 in monkey H. In Exp-II for PRR and PMd it was 19.1 and 15.7 in monkey K, and 10.6 and 11.6 in monkey H.

## Correlation method for testing object-centered and egocentric reference frames hypotheses based on single units' firing rates

At the single unit level, we distinguished object-centered and egocentric reference frames by comparing each neuron's spatial selectivity profile for the five cue stimulus positions in trials when an object was presented on the left side with the trials when the corresponding object was presented on the right side. During visual memory, we compared the reference object positions, irrespective of reach object position. Vice versa, during movement planning we compared the reach object positions, irrespective of reference object position. The object-centered hypothesis predicts object-left and object-right selectivity profiles to be identical in shape, but shifted relative to each other in egocentric screen space (Fig. 2a–c, left). The egocentric hypothesis instead predicts the selectivity profiles to be identical only in the range of object compartments that overlap in egocentric screen space (Fig. 2a–c, right). For each cell, we measured the similarity of the two selectivity profiles to quantify how well either hypothesis can explain the response of the neuron. We used the Pearson's linear correlation coefficient, similar to[82], between each pair of the selectivity profiles, i.e., reference object-left versus -right; reach object-left versus -right. The Pearson's correlation coefficient between vectors $X$ and $Y$ of length $n$, was calculated as follows:

$$Corr(X,Y) = \frac{\sum_{i=1}^{n}(X_i - \bar{X})(Y_i - \bar{Y})}{\sqrt{\sum_{i=1}^{n}(X_i - \bar{X})^2 \sum_{j=1}^{n}(Y_j - \bar{Y})^2}} \qquad (1)$$

For variance-stabilization, the bounded correlation coefficients were subjected to Fisher's z-transformation (inverse hyperbolic tangent function) before using them for any further analysis and statistical tests. For brevity, we refer to the Fisher z-transformed correlation coefficients as "correlation coefficients". The inverse hyperbolic tangent of the correlation coefficients was calculated using the following formula:

$$tanh^{-1}(x) = \frac{1}{2}\log\left(\frac{1+x}{1-x}\right) \qquad (2)$$

Since the correlation coefficient is insensitive to linear scaling, the similarity between the shapes of the selectivity profiles is quantified independent of potential gain modulation effects on the neural firing rates. This means, if object location modulates responses to all cue positions by an equal factor, without affecting relative response strengths for different on-the-object cue positions, the correlation method considers this outcome to be in accordance with the object-centered hypothesis. In contrast, our complementary decoding method (see below) is sensitive to gain effects on neural activity.

For each cell and pairwise selectivity profile, the correlation coefficient was calculated twice, once in each reference frame. The object-centered correlation (objCorr) is the correlation coefficient between the samples of the object-left and -right selectivity profiles with corresponding object-centered cue locations. (Fig. 2a, left, green arrows show points with equivalent object-centered location). The egocentric correlation (egoCorr) is the correlation coefficient between samples of the object-left and -right selectivity profiles with corresponding egocentric locations. (Fig. 2a, right, dark red arrows show points with equivalent egocentric location). Each of the correlation values provides a measure that determines the validity of the corresponding hypothesis for each cell's activity profile.

The difference objCorr-egoCorr between them indicates the neuron's preference of reference frame. We will refer to this difference as Position Invariance (PI). Positive values indicate selectivity profile of the neuron was better explained by the object-centered hypothesis (= invariance with respect to egocentric location), while negative values point towards the egocentric hypothesis (= invariance with respect to object-centered location). Because the number of samples (task conditions) for calculating objCorr is larger than egoCorr (5 vs 4), one concern might be that this could bias the objCorr towards larger values and consequently yield in positive PIs. In a control analysis, we calculated objCorr by randomly selecting 4 boxes on the object, and averaged across all possible selections. Then compare the PI value with unequal sample size (Supplementary Fig. 7). We did not see significant difference between the PIs when tested for two time intervals: last 300 ms of memory period and last 300 ms of planning period.

In order to reduce effect of outliers in trial-by-trial firing rates, we estimated the mean firing rate for every unit with a bootstrap method (200x resampling; bootstrp() function in MATLAB) and the PI was calculated for every bootstrap sample. In addition, for every bootstrap sample, the null distribution of PI was estimated by randomly shuffling trials (trials used to generate this bootstrap sample) 100 times, across all task conditions, and the average null-PI was subtracted from the corresponding PI.

## Visual memory vs movement planning reference frame

To study changes in the preferred reference frame across time, for each cell PI was calculated in 300 ms windows, sliding by 50 ms, extending from 600 ms before until 600 ms after reach object onset (data was aligned to reach object onset). In all time bins earlier than reach object onset (up to bin [−150 150] ms), selectivity profiles were computed for cue relative to reference object (memory period); after reach object onset, selectivity profiles were computed for target relative to reach object (movement planning period). The time bin centered on reach object onset ([−150 150]ms) was included in both memory and movement planning periods.

## Scaling of the spatial selectivity to the object size in Exp-II

Exp-II aimed at further characterizing object-centered encoding. Therefore, we analyzed firing rates only during the late memory period when neurons have reached their sustained activity and predominant position-invariant (object-centered) preference was expected based on the results of Exp-I. The object-centered hypothesis predicts that selectivity profiles should scale with the object size in the egocentric screen space (Fig. 4a, left), independent of its location, meaning similar

object-centered selectivity profiles in all four conditions of object size and location. The egocentric hypothesis predicts that selectivity profiles for small object size should resample part of the selectivity profile for large object size in the egocentric screen space (Fig. 4a, right). To test this, we quantified two correlation coefficients for every neuron: alloCorr was the correlation coefficient between a neuron's activity in ten conditions (5 on-the-object cue positions times 2 reference object locations) in long-object trials and their corresponding conditions (with identical object-centered cue location) in short-object trials (2 sets of 10 green arrows in Supplementary Fig. 1a). egoCorr was the correlation coefficient between neuron's activity in six conditions (3 on-object cue positions times 2 reference object locations) in long-object trials and their corresponding conditions (with identical egocentric locations) in short-object trials (2 sets of 6 red arrows in Supplementary Fig. 1b). To account for both size and position invariance at the same time, alloCorr and egoCorr were calculated across all possible pairing of the firing rates between the short and long trials, regardless of object location. We calculated alloCorr-egoCorr to quantify how far a neuron complies with the prediction of object-centered Position and Size Invariant (PSI) encoding, and refer to this measure as PSI. In the same way as in Exp-I, PSI for every cell was bootstrapped and a shuffle predictor subtracted. We did not see significant effect of unequal number of samples between alloCorr and egoCorr on the PSI (see explanation above and Supplementary Fig. 8).

### Decoding population activity

As an additional population-level analysis, we used a cross-conditional decoding approach: A 5-way classifier for decoding cue or target position on the object was trained with data from one condition (e.g. reference object located left on screen) and its performance was tested on data from another condition (reference object-right). For Exp-I, there are two conditions, object-left and object-right, in the memory and planning period, respectively. A decoder that is only trained on trials of the object-left condition but accurately predicts the position in object-right condition trials (left-right generalization) suggests a predominant position-invariant (object-centered) encoding in the data. This will show as a confusion matrix where only the main diagonal is populated (Fig. 3d). On the other hand, a systematic offset in the predicted labels, e.g., object-right trials at positions 1, 2, 3, 4 (counting the boxes from left to right) are decoded as positions 2, 3, 4, 5 by the object-left classifier, is indicative of an egocentric encoding, because these positions correspond to the same egocentric locations. Such pattern would show as a confusion matrix populated along the secondary diagonal. In Exp-II, there are four conditions, long-object-left, long-object-right, short-object-left, short-object-right, and each pair has to be compared independently. To quantify the size-invariance of the neural population, we compared each long- with each short-object condition, resulting in four comparisons.

For the decoding analysis, we included only units for which we recorded at least 15 trials per cue/target box-on-the-object. This enabled us to select a random subset of 15 trials per box-on-the-object for training from each unit. The trial-by-trial average firing rates of the units were used to form the 75 feature vectors ( = 15 trials × 5 boxes on the object) of size [1 x number of units], which provides a matrix of size [75 x number of units] for training the decoder. Since most neurons were not recorded simultaneously, with random trial matching across neurons, we could include the whole dataset in the decoding analysis and create feature vectors in the high dimensional neural space that included all neurons (all neurons that have at least 15 trials per task conditions). Feature vectors for testing were built similarly by selecting a random subset of 15 trials (or more, if the minimum number of trial per box-on-the-object across all neurons was more than 15) from the same units. Feature vectors for testing included same number of trials per box-on-the-object for all neurons. This randomization was repeated 1000 times, each time choosing random subsets of trials from

each unit. For each repetition, an independent classifier was trained. We used an error-correcting output codes (ECOC) model with 10 binary linear support vector machines as the classifier (MATLAB function: fitcecoc()). To get a time-continuous estimate, we repeated the analysis, each time shifting the window by 50 ms. During the visual memory period, the condition of each trial was defined by the reference object, during the movement planning period it was defined by the reach object. For the time windows that overlapped with both periods (around reach object onset), the condition was defined by the period with the most overlap. For the window centered on the reach object onset, both definitions where used.

In order to quantify the dominance of a reference frame, we calculate the difference $p_{obj} - p_{ego}$ in percentage of trials decoded in accordance with the object-centered hypothesis and percentage of trials decoded in accordance with the egocentric hypothesis. This measure will be referred to as Reference Frame Difference (RFD). A value of 1 shows complete object-centered encoding, while a value of −1 shows complete egocentric encoding. For this calculation, we only consider the test trials to positions that have an egocentric equivalent in the training condition. For example, we cannot include test trials to the right-most position when the object is on the right, because this location does not exist in the object-left condition (Fig. 1a and Fig. 3d). The number of eligible positions depends on the conditions that are being compared: In Exp-I, four positions overlap (Fig. 3d) while for comparisons of short and long objects in Exp-II, two positions can be considered (three positions overlap, but one of these is the same in both reference frames and hence does not help to distinguish them, Supplementary Fig. 1).

To test for a significant difference from zero, we calculate the RFD for each of the 1000 randomizations and compute the percentage of values lying below and above 0. If one of these percentages is less than half the alpha criterion, it is considered significant.

In all analysis where results of statistical tests were compared across multiple time bins, the significance level was adjusted to account for multiple comparisons by correcting for the false-discovery rate, allowing a proportion of false positives of less than 5%.

We used Gramm[83] MATLAB toolbox for plotting some of the figures.

### Animal implantation and neural recordings

The procedures for animal preparation and neural recordings were described previously[25] and are here summarized in the context of Supplementary Note 2 and Supplementary Fig. 10, for completeness.

Both animals were housed in social groups with one or two male conspecifics in facilities of the German Primate Center. The facilities provide cage sizes exceeding the requirements by German and European regulations, and access to an enriched environment including wooden structures and various toys. All procedures have been approved by the responsible regional government office [Niedersächsisches Landesamt für Verbraucherschutz und Lebensmittelsicherheit (LAVES)] under permit numbers 3392 42502-04-13/1100 and comply with German Law and the European Directive 2010/63/EU regulating use of animals in research.

### Reporting summary

Further information on research design is available in the Nature Portfolio Reporting Summary linked to this article.

## Data availability

Data for reproducing figures and statistics can be found here: https://doi.org/10.25625/EO4DDY Source data are provided with this paper.

## Code availability

Code for reproducing figures and statistics can be found here: https://doi.org/10.25625/EO4DDY.

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

## Acknowledgements

We thank Sina Plümer for help with data collection and technical support, Klaus Heisig for help with setup maintenance, Leonore Burchardt for help with animal training and Dirk Prüße for technical support. This work was supported by and benefited from the State of Lower Saxony (grant VWZN2563), the European Commission in the context of the Plan4Act consortium (http://plan4act-project.eu; EC-H2020-FET-PROACT-16732266), and the German Research Foundation in the context of the Collaborative Research Center Cognition of Interaction (DFG SFB-1528).

## Author contributions

B.T. and A.G. designed and implemented the experiment, and collected the data. B.T. and O.F. analyzed the data. B.T., O.F. and A.G. discussed the results and wrote the paper.

## Competing interests

The authors declare no competing interests.
