## [Peer Review File · Nature Communications]

Position- and scale-invariant object-centered spatial localization in monkey frontoparietal cortex dynamically adapts to cognitive demandREVIEWER COMMENTS

Reviewer #1 (Remarks to the Author):

This study examines a very timely and relevant topic in an understudied field of research (compared to research on egocentric reference frames and saccadic eye movements). The results are novel and important as they show that allocentric coding is not predominantly processed in the ventral visual stream, as some models hypothesize (e.g. Neggers et al., 2006). The results show that allocentric coding for reaching is processed in reach-related areas of the dorsal visual stream (PRR and PMd) and that neurons in one and the same brain area represent reach targets in both egocentric and allocentric reference frames. Importantly, this spatial coding pattern is not static but coding preferences change with task demands. Such dynamic changes in spatial representations are inconsistent with the view that the visual system is segregated into two functional pathways for egocentric and allocentric encoding in the dorsal and ventral visual stream, respectively.

To investigate the neural underpinnings of allocentric coding for reaching the authors applied a clever and well thought-through experimental design. The two manipulations (change in object position and change in object size) are straight forward and task scholarly designed. The results of Experiment 2 support the results of Experiment 1 which strengthen the overall findings and conclusions. The statistical analyses are sound and include innovative approaches to compare egocentric and allocentric coding schemes and the results of the two experiments. My overall impression is very positive and I am convinced that this piece of work would make a valuable contribution to the field and is appealing to a larger readership. I have several, largely minor comments to further improve the manuscript.

1. I found the paper easy to read and follow. However, given the overall structure of the paper with the methods at the end I missed some information in the introduction or result sections that would have helped to better follow the results. In particular, the hypotheses (object-centered hypothesis, egocentric hypothesis, recruitment hypothesis) should be explained at the very first appearance in the text.
2. One of the main messages of the paper is that the predominant frame of reference in PRR and PMd shifts according to cognitive demands that change over the course of the trial. This construct 'cognitive demands' is never really defined. It would be helpful to explicitly define the term in the context of the current task and explain the different demands associated with the different trial phases.
3. Previous studies on egocentric and allocentric coding mainly looked at saccadic eye movements. A very recent study of the Crawford lab proposed a convolutional network model for egocentric/allocentric integration and suggested that this model is not exclusive to eye movements but could also explain other complex visuomotor behaviors. A discussion of the present results with respect to this model would stimulate further discussion in the field.

Abedi Khoozani, P., Bharmauria, V., Schütz, A., Wildes, R. P., & Crawford, J. D. (2022). Integration of allocentric and egocentric visual information in a convolutional/multilayer perceptron network model of goal-directed gaze shifts. *Cerebral Cortex Communications*, (3), tgac026.

In addition, a similar shift from more allocentric coding during the memory period to more egocentric coding during action preparation has also been reported in the saccade system. This suggests that such a dynamic change in spatial coding patterns is not restricted to one sensorimotor system but rather a more general mechanism. Therefore, it would be beneficial to expand the discussion a bit into this direction. See for example:

Bharmauria, V., Sajad, A., Li, J., Yan, X., Wang, H., & Crawford, J. D. (2020). Integration of eye-centered and landmark-centered codes in frontal eye field gaze responses. *Cerebral Cortex*, 30(9), 4995-5013.

4. Given the rich data set, I was wondering if the authors could say a bit more about the cortical timeline of activation in PRR and PMd, similar to their previous work (e.g., Westendorff et al., 2010, *J Neurosci*). This could add an additional piece of exciting new information to better understand the integration of different spatial coding schemes.

5. Page 9: Please add a reference to the Hartigan's dip test.

6. Page 9: Please explain the rationale of training the classifier for trials where the object was on one side and tested on trials where the object was on the other side (in contrast to having the object at any other location).

7. Please provide the exact p-values if the value is larger than $p > 0.001$ and add an effect size measure.

8. Page 17: Why were the times of holding the target different for monkey H (220 ms) and monkey K (300 ms)?

9. Page 19: I wasn't able to follow the description of the behavioral success rate. Please clarify.

10. Page 25: The long section about animal implantation and neural recordings could be deleted because it is explained in Westendorff et al., 2010.

11. The figures of the main text are very well prepared. There are several small inconsistencies (e.g. spacing between text and image; choice of font size) in the figures of the supplement that could be improved.

Reviewer #2 (Remarks to the Author):

This manuscript describes experiments probing the reference frames used to encode target locations in two dorsal stream reaching planning areas: dorsal premotor cortex and the parietal reach region. The experiments are designed to specifically examine the presence of object-centered coding of location, in comparison to the egocentric frames that have previously been used to describe activity related to spatial location in these areas. Although there are some noteworthy results, there are also multiple issues with the methodology and presentation of these results that make it difficult to assess their overall significance to the field and related fields.

The manuscript is generally not well-written. In addition to many instances of awkward phrasings (only some of which are highlighted below), there are multiple grammatical errors (tense, number (plural vs singular)), and also some typographical errors. In some cases these obscure the meaning of key passages, which will distract and/or confuse many readers.

In general, the number of neurons analyzed is low by today's standards. In Exp II, for example, only ~20 neurons were analyzed for monkey K in each area. In my opinion, this essentially makes Exp II a single animal study. At present all that can be said is that a small number of neurons have been identified that appear to encode locations in a manner consistent with an object-centered representation. At the population level, the results appear to show that at certain times activity is more strongly influenced by a target's location on an object than on its egocentric location. While interesting, this could simply reflect activity in ventral stream areas to which these areas are connected, rather than a fundamental property of dorsal stream areas themselves.

Key terms and concepts are not well-defined. Among these are the reference frames that are the focus of the manuscript. For example, allocentric (world-centered) and object-centered coordinates are not synonymous but are treated as if they are at several points in the manuscript. In addition, what is referred to as 'egocentric' (a generic term for a reference frame anchored to some unspecified body part) is more precisely 'not object-centered'. That is, since the position of the trunk/head/arm in space was not varied with respect to the reach targets, it is impossible to distinguish between a true egocentric representation and an allocentric one.

Another term which is not defined very precisely is 'scale invariance'. In these experiments, the authors kept the overall size of the object the same in the vertical dimension but scaled it horizontally, which changed the spacing between targets. However the object itself could have also been scaled vertically and horizontally, which would change the spacing between targets in world coordinates more generally. As far as I can tell the rationale behind scaling in one vs two dimensions is never discussed, nor is its significance for understanding spatial representations in these areas.

Given these issues it is difficult to grasp the true significance of the results.

Below are some specific suggestions for improving the presentation.

Introduction

Line 53: I don't believe 'subject-independent' is an adequate synonym for allocentric if that is what this parenthetical statement is implying.

Lines 56-57: Awkward phrasing, recommend rewriting.

Line 83: Doesn't sound like this result demonstrates strictly allocentric coding, it sounds more context-dependent.

Line 118: for clarity suggest rephrasing to something like: '... encode visual cues and reach targets in object-centered as well as egocentric reference frames, with the predominating reference frame in both areas being dynamically adjusted based on cognitive demands'.

Results

Line 138: for clarify: '... shifted horizontally ...'

Line 140: for clarity, suggest adding that in this experiment position could be shifted horizontally or vertically.

Line 150: would recommend sticking with 'egocentric' rather 'body-centered' to avoid confusion with a 'trunk-fixed' reference frame, which was not strictly examined.

Line 159: '...encode the locations of visual cues and reach goals ...'

Line 165: 'analyses'

Line 166-168: See general comments regarding the number of neurons analyzed.

Lines 169-172: Although idealized neurons are straightforward, real neurons often show partially shifted and/or scaled response profiles. As a result it is important to point out up front what 'consistent with encoding of cue location in an object centered reference frame' really means. In this example, if the profiles were plotted in object centered coordinates they would likely appear partially shifted and scaled.

The possibility of mixed encoding comes up later but should be discussed prior to presenting real responses and describing them as resembling object-centered or egocentric. Being clear on this point is imperative since the authors later state that they didn't discretely categorize responses as object-centered or egocentric.

Lines 177-179: I read the sentence beginning with 'Neural response strengths ...' several times and can't decipher its meaning.

Lines 182-184: As indicated above, these statements should come earlier and 'mixed encoding' should be defined.

Lines 186-188: Without showing one of these examples it will be very difficult for readers to understand these statements. Also, 'linear selectivity profile'?

Lines 191-194: Again, the language here will likely confuse many readers. Don't the authors mean to say that such gradual tendencies 'were quantified to determine whether they differed' rather than 'can be quantified and might differ'?

Lines 209-218: It would likely be more helpful to readers to have the all possible scenarios presented first, and in a more fully fleshed manner, followed by a description of the real data. For example, 'recruitment' hypothesis: bimodal distribution with lobes pointing upper right and lower left; 'recoding'

hypothesis: unimodal distribution pointing lower right. It might also be useful to entertain the idea of a 'reverse recruitment' scenario (unimodal pointing upper left), if only to emphasize that this non-intuitive scenario is not matched by very many neurons.

Lines 247-249: The description of the statistical analysis here is adequate. Assuming a t-test was used, the exact type of t-test, degrees of freedom, and t-statistic should all be reported.

Lines 255-257: The 'real world' examples of scaling described here involves scaling in two dimensions and are therefore not good examples of the type of scaling used here, which was along a single dimension. This might seem like nitpicking but readers may wonder about the relevance of the authors chosen scale manipulation so it would be good to justify it with better examples.

Lines 267-271: See general comments regarding the number of neurons analyzed.

Lines 282-284: To avoid confusion, this statement should come earlier, when the experiment is first being discussed.

Lines 284-285: I don't understand what this statement is referring to.

Line 291: Also an inadequate description of a statistical analysis. Please use proper punctuation and exponential notation.

Figures

Figure 2b: Suggest labeling these with some identifying information such as cell numbers, areas, etc.

Figure 2c: I found it very difficult to distinguish among the color gradations in this part of the figure which then made it difficult distinguish which line belonged to which object location.

Figure 3b: I found this plot to be unnecessarily confusing. The authors essentially transformed the data from Cartesian to polar coordinates to compute an angular histogram but retained the Cartesian coordinates on the axes. Why? Suggest removing the Cartesian axes and just stating that the indices were converted to polar coordinates for this analysis. Incidentally, since they have done this transformation they could use the distance from the origin ('r') to potentially remove neurons that are clearly not biased toward any quadrant (which appear to be many) and are likely influencing their statistical analyses.

Figure 3c (caption): Typographical error: 'zerp' should be 'zero'.

Figure 4c: As with Figure 2c, the color gradations are too difficult to distinguish.

Discussion

Given the issues raised in the general comments (particularly regarding the frames of reference) it is difficult to comment on this section of the manuscript.

Reviewer #3 (Remarks to the Author):

The brain needs and employs multiple reference frames to encode the location of objects and events around. Taghizadeh and colleagues' report is a timely and interesting study which show original data which are crucial to understand the duality of egocentric and object-centered representation of action targets. The paper is clear and well written, the study is well designed and the results properly interpreted. Few things are still to be discussed and some points to be better clarified.

Something seems a bit misleading in the title: « ...dynamically adapt to task demand ». Actually the task is the same for all presented trials, that is to identify and remember the target location of an action to be executed at the given signal. What if the goal was different and the subsequent action absent? Would the dynamic of the activation be the same or different during the memory phase? This said, certainly the data show that there's a shift from an object-centered spatial encoding during memory to an egocentric one during planning (beautiful result), but both are elicited by the same task demand at every trial.

The object shift manipulation from reference to target position changes the location of the object horizontally, and it seems on the same azimuth on the screen. Why did the authors make this choice? Wouldn't the object-centered reference hypothesis predict similar results even when changing the position of the object vertically from reference to target? Could the constant azimuth itself elicit or at least facilitating the activation shift taken as proxy of the object-centered reference?

A further discussion point is which signal triggers the shift from object-centered to egocentric. Unless I'm missing something, no information is given concerning the relative temporal dynamics of the activity in PRR and PMd. Is one leading the other? Are they both receiving signals from elsewhere entailing a shift?

Reviewer #4 (Remarks to the Author):

The paper by Taghizadeh et al. investigates whether single neurons in premotor dorsal and in parietal cortex encode cue and reach target in reaching tasks in object-centered and/or egocentric reference frames of reference. They find that the predominant frame of reference shifts according to cognitive demands in different epochs of the behavioral task. Specifically, in both PMd and PRR the object-centered encoding predominated during visual memory while egocentric encoding predominated during movement planning. The Authors also provide the first electrophysiology evidence for size-invariant spatial selectivity in both cortical regions.

The paper is well written, figures are clear and compelling and data are reliable and analyzed in several ways. I only have a couple of main concerns and a minor one.

Main concerns:

1) Recording site: The Authors report the coordinates of recording chambers. This roughly indicates the cortical region where they record from, but the reader does not know where they exactly record from. For instance, they say they record from PRR, and claimed that “sustained direction-selective neural responses during center-out reach planning (memory period) served as physiological signature to confirm the region of interest”. But neurons active during planning of reaching are present in both SPL and IPL as well as in both walls of the intraparietal sulcus. Where did they actually pick-up the neuronal activity described in the paper? In short, the recording sites must be shown.

2) The egocentric frame of reference repeatedly reported in the paper could be an eye-centered, head-centered, or body-centered frame of reference. Although the term ‘egocentric’ used in the paper in comparison with ‘allocentric’ can be accepted, because the used task does not allow to discriminate eye-centered, head-centered, and body-centered frame of reference, in my opinion the point of different egocentric frames of reference should be recalled and discussed in the paper, given that in both parietal and premotor cortex, neurons encoding space in these different frames of reference are reported to be present.

Minor concern:

If I’ve correctly understood, the ‘white square’ reported to be used as hand fixation stimulus is shown, in Fig. 1, as a small black square. This raises some confusion. Please modify.

Signed:

Patrizia Fattori

REVIEWER COMMENTS

Reviewer #1 (Remarks to the Author):

This study examines a very timely and relevant topic in an understudied field of research (compared to research on egocentric reference frames and saccadic eye movements). The results are novel and important as they show that allocentric coding is not predominantly processed in the ventral visual stream, as some models hypothesize (e.g. Neggers et al., 2006). The results show that allocentric coding for reaching is processed in reach-related areas of the dorsal visual stream (PRR and PMd) and that neurons in one and the same brain area represent reach targets in both egocentric and allocentric reference frames. Importantly, this spatial coding pattern is not static but coding preferences change with task demands. Such dynamic changes in spatial representations are inconsistent with the view that the visual system is segregated into two functional pathways for egocentric and allocentric encoding in the dorsal and ventral visual stream, respectively.

To investigate the neural underpinnings of allocentric coding for reaching the authors applied a clever and well thought-through experimental design. The two manipulations (change in object position and change in object size) are straight forward and task scholarly designed. The results of Experiment 2 support the results of Experiment 1 which strengthen the overall findings and conclusions. The statistical analyses are sound and include innovative approaches to compare egocentric and allocentric coding schemes and the results of the two experiments. My overall impression is very positive and I am convinced that this piece of work would make a valuable contribution to the field and is appealing to a larger readership.

We thank the reviewer for this encouraging feedback.

I have several, largely minor comments to further improve the manuscript.

1. I found the paper easy to read and follow. However, given the overall structure of the paper with the methods at the end I missed some information in the introduction or result sections that would had helped to better follow the results. In particular, the hypotheses (object-centered hypothesis, egocentric hypothesis, recruitment hypothesis) should be explained at the very first appearance in the text.

We agree and followed this recommendation in three ways:

First, we added explanations/working definitions for the most important concepts (e.g. allocentricity) to the introduction section, lines 63-68. Second, we added the following text about the object-centered and egocentric hypotheses after their first appearance in the introduction section, lines 130-134:

“Animals had to identify reach goals relative to a visual object where the object could appear randomly at two potential positions on the screen. We asked if neural selectivity patterns are best explained as a function of the target location on the object (object-centered hypothesis), or the target location relative to the body of the animal (egocentric hypothesis).”

Third, for the testing of the recruitment against the re-coding hypotheses, we added explanations of both hypotheses to the main text, added hypothetical plots to Fig. 3, and moved the detailed explanation of our angular distribution analysis from the methods to the result section, lines 236-262.

2. One of the main messages of the paper is that the predominant frame of reference in PRR and PMd shifts according to cognitive demands that change over the course of the trial. This construct 'cognitive demands' is never really defined. It would be helpful to explicitly define the term in the context of the current task and explain the different demands associated with the different trial phases.

We added the following clarification to the introduction section, lines 77-82.

"This suggests that spatial processing needs to be cognitively controlled so that those spatial parameters are available at different stages of action preparation that are relevant in the respective moment. In the earlier phase, when deciding where along the stick to pick it up, object-centered encoding relative to the stick is most relevant. Later, for planning the physical movement of the hand, egocentric postural signals might be more relevant."

3. Previous studies on egocentric and allocentric coding mainly looked at saccadic eye movements. A very recent study of the Crawford lab proposed a convolutional network model for egocentric/allocentric integration and suggested that this model is not exclusive to eye movements but could also explain other complex visuomotor behaviors. A discussion of the present results with respect to this model would stimulate further discussion in the field. Abedi Khoozani, P., Bharmauria, V., Schütz, A., Wildes, R. P., & Crawford, J. D. (2022). Integration of allocentric and egocentric visual information in a convolutional/multilayer perceptron network model of goal-directed gaze shifts. *Cerebral Cortex Communications*, (3), tgac026.

In addition, a similar shift from more allocentric coding during the memory period to more egocentric coding during action preparation has also been reported in the saccade system. This suggests that such a dynamic change in spatial coding patterns is not restricted to one sensorimotor system but rather a more general mechanism. Therefore, it would be beneficial to expand the discussion a bit into this direction. See for example:

Bharmauria, V., Sajad, A., Li, J., Yan, X., Wang, H., & Crawford, J. D. (2020). Integration of eye-centered and landmark-centered codes in frontal eye field gaze responses. *Cerebral Cortex*, 30(9), 4995-5013.

Thank you. We added the references to our discussions about other existing computational models, about the relationship between eye and hand movements in allocentric encoding, and about dynamic shifts in reference frames.

4. Given the rich data set, I was wondering if the authors could say a bit more about the cortical timeline of activation in PRR and PMd, similar to their previous work (e.g., Westendorff et al., 2010, *J Neurosci*). This could add an additional piece of exciting new information to better understand the integration of different spatial coding schemes.

We agree with the reviewer. We would love to draw strong conclusions about relative latencies between PMd and PRR, but are careful, given the diverse findings between the PI and RFD measure (Fig. 3a/d). Below, at the end of this rebuttal, we present additional analyses based on the correlation coefficients (PI measure) in more detail. These results do not support the view that the shift in reference frame occurs first in PMd, different to what the RFD results suggest. We added the following comment to the Discussion section to give a balanced account of this situation (lines 416-423):

"In our decoding analyses, PMd reflected the allo-to-ego transition earlier than PRR (Fig. 3c), while correlation-based measures (Fig. 3a; plus an alternative measure based on the objCorr/egoCorr ratio; data not shown), did not indicate a latency difference between

both areas. This means, if there is a latency difference at all, then PMd leads the shift in reference frame. Such finding would support earlier conclusions that frontal areas lead parietal areas in determining motor goals when tasks require a spatial remapping²⁷, and the more general notion that frontal lobe exerts cognitive control over posterior regions of the brain."

Please note, since this question was also raised by reviewer #3, we added a section at the end of this rebuttal letter with more details about how we assessed difference between temporal dynamics of PRR and PMd, including a third method complementing the two methods in the main manuscript. Please see "Relative temporal dynamics between the areas" at the end of this letter for details.

5. Page 9: Please add a reference to the Hartigan's dip test.

Thank you for pointing this out. The reference is now added (ref. 49)

6. Page 9: Please explain the rationale of training the classifier for trials where the object was on one side and tested on trials where the object was on the other side (in contrast to having the object at any other location).

We are not clear what the reviewer means by "any other location". We hope that elaborating on the decoding strategy helps to clarify open questions: In the cross-conditional classification, classifiers were trained to distinguish the five positions on the object when they were trained only on trials where the object was on one side. We then tested classifier performance on all other trials which were not included in the training set. Training and testing was done independently in every time bin. In the time bins of the memory period, the five locations of the cue on the reference object were classified; in time bins of the planning period, after appearance of the reach object, the five locations of the reach target position (not visible) on the reach object were classified. The object (reference or reach object) could be presented on the screen either with an offset to the left or to the right of the midline. Therefore, when the classifiers were trained for left (right) object trials, the only other location in which their classification performance could be tested in the cross-conditional approach were the trials when the object was presented on the right (left) side.

7. Please provide the exact p-values if the value is larger than $p > 0.001$ and add an effect size measure.

Thank you. We reported the p-values where $p > 0.001$, line 330.

8. Page 17: Why were the times of holding the target different for monkey H (220 ms) and monkey K (300 ms)?

We would have preferred to keep the exact same timing for both animals but monkey H's performance was better when reward was delivered to him with slightly shorter "holding" delay. We consider the difference irrelevant for the aims of our study since either of the two "hold" times is long enough to require the animals to rest the hand briefly on the target position, and since the time is at the very end of the trial (after the reach and clearly out of our desired analysis window). We hence opted for the slight variation in the task parameter to gain higher performance in the second animal, aka more successful trials per session, in order to optimize statistical power.

9. Page 19: I wasn't able to follow the description of the behavioral success rate. Please clarify.

We apologize if our description of success rate was unclear. We expanded it in the following way (line 604-609):

“Behavioral success rate was calculated as percentage of correctly performed reaches relative to the number of trials that the monkey initiated and in which it attempted to perform the task by initiating a reach movement. Successfully initiated but unsuccessfully completed trials include cases of belated start of the reach, touching the wrong box on the object, touching out of the predefined tolerance window around the target box, touching the target box but not holding the position long enough, or breaking eye fixation..”

We excluded trials where the monkeys briefly started the trial but soon after broke eye or hand fixation before even attempting to perform the reach task. This typically happened in phases of low motivation, e.g., towards the end of the session or on certain days. Therefore, we feel that they are not well suited to characterize the task performance per se but rather the motivational state of the animal.

10. Page 25: The long section about animal implantation and neural recordings could be deleted because it is explained in Westendorff et al., 2010.

We agree that such redundancy is not necessary. At the same time, some readers prefer to have the key information readily at hand. We now moved the section "Animal implantation and neural recording" to the supplementary information and, following a request by reviewer #4, combined it there with additional detail on the recording sites, which is important information specific to this study, not available from previous methods descriptions..

11. The figures of the main text are very well prepared. There are several small inconsistencies (e.g. spacing between text and image; choice of font size) in the figures of the supplement that could be improved.

Apologies. We revised all figures to improve the quality, including supplementary figures.

Reviewer #2 (Remarks to the Author):

This manuscript describes experiments probing the reference frames used to encode target locations in two dorsal stream reaching planning areas: dorsal premotor cortex and the parietal reach region. The experiments are designed to specifically examine the presence of object-centered coding of location, in comparison to the egocentric frames that have previously been used to describe activity related to spatial location in these areas. Although there are some noteworthy results, there are also multiple issues with the methodology and presentation of these results that make it difficult to assess their overall significance to the field and related fields.

The manuscript is generally not well-written. In addition to many instances of awkward phrasings (only some of which are highlighted below), there are multiple grammatical errors (tense, number (plural vs singular)), and also some typographical errors. In some cases these obscure the meaning of key passages, which will distract and/or confuse many readers.

We thank the reviewer for pointing out language issues and for useful suggestions for improved phrasing. We tried our best to fix grammar mistakes and poor phrasing in the revised manuscript.

In general, the number of neurons analyzed is low by today's standards. In Exp II, for example, only ~20 neurons were analyzed for monkey K in each area. In my opinion, this essentially makes Exp II a single animal study. At present all that can be said is that a small number of neurons have been identified that appear to encode locations in a manner consistent with an object-centered representation.

We analyzed and presented the two monkeys individually in the supplementary figures and only reported the results in our manuscript which were significant at the population level and consistent across both animals. This is also the case for Exp-II, where both the correlation- and the decoding-based analyses yield significant object-centered encoding. In so far, we disagree with the notion of only showing a few suggestive example neurons or dealing with an N=1 study. In general, lack of statistical power would be more of a concern when reporting null results (lack of effect), which we do not do, or when having a high risk of a biased sample. We do not see a risk of such bias since we did not preselect the neurons in any task-related way, but instead included all neurons that were well isolated and for which we recorded enough trials. In summary, since we find effects across different methods and across different animals at the neural population level, in some cases even with "only" several dozens of neurons, we can assume that the effect must be quite robust.

At the population level, the results appear to show that at certain times activity is more strongly influenced by a target's location on an object than on its egocentric location. While interesting, this could simply reflect activity in ventral stream areas to which these areas are connected, rather than a fundamental property of dorsal stream areas themselves.

First, as was discussed in the manuscript (lines 486-502), the dual visual pathway model does not predict fast temporal dynamics of interaction between the ventral and dorsal stream areas. Behavioral studies predict longer latencies (in the order of seconds) for interaction between the two pathways, whereas our data show object-centered encoding in the dorsal stream with much shorter latency (starting around the time of disappearance of the visual cue). To the best of our knowledge, there is no electrophysiology study to directly test the temporal aspects of neuronal interactions between the two streams in the context of allocentric encoding. Second, we observed mixed reference frames in our data as they result from computational mechanisms of reference frame transformations within an area (Avillac et al., 2005; Brozovic et al., 2007; Deneve et al., 2001; Deneve and Pouget, 2003), as also discussed on the manuscript. Based on these arguments, we consider it most likely that PRR and PMd take an active role in object-centered encoding of reach targets, and not just "echo" other parts of the brain.

On a more general note: In recurrent networks like the brain, probing of any node will reflect local characteristics as well as feedback characteristics with other nodes. This is true for all neurophysiological studies in highly connected brain areas and not specific to our study. If a certain property, like object-centered encoding, predominates during a specific task in a brain area, on which basis would one call it "fundamental" in one case and epiphenomenological in the other? Even for (seemingly) "causal" methods like local inactivation, such distinction is extremely difficult due to the feedback mechanisms in recurrent networks. If dorsal stream activity reflects allocentric information about reach targets in certain contexts, while its "true" function is egocentric encoding, then this is still an important observation in order to understand its role in the network, since connected nodes in the network will be influenced by its output either way.

Key terms and concepts are not well-defined. Among these are the reference frames that are the focus of the manuscript. For example, allocentric (world-centered) and object-centered

coordinates are not synonymous but are treated as if they are at several points in the manuscript.

We thank the reviewer for pointing out the challenges resulting from different nomenclatures used in the literature. We consider object-centered encoding as an example of allocentric encoding schemes, head-centered encoding as an example of egocentric schemes. We added a short paragraph about reference frames, including our working definition of “object-centered”, to the introduction section, lines 63-66.

In addition, what is referred to as ‘egocentric’ (a generic term for a reference frame anchored to some unspecified body part) is more precisely ‘not object-centered’. That is, since the position of the trunk/head/arm in space was not varied with respect to the reach targets, it is impossible to distinguish between a true egocentric representation and an allocentric one.

We agree and apologize if this logic did not become clear enough in the previous version. To make it more evident, we added explanations to our use of “egocentric” to the introduction (lines 66-68) and methods sections (lines 507-511). Given evidence from previous studies of other labs, we will still refer to the non-object-centered encoding as egocentric rather than interpreting everything as allocentric (discussion section, lines 383-387).

Another term which is not defined very precisely is ‘scale invariance’. In these experiments, the authors kept the overall size of the object the same in the vertical dimension but scaled it horizontally, which changed the spacing between targets. However the object itself could have also been scaled vertically and horizontally, which would change the spacing between targets in world coordinates more generally. As far as I can tell the rationale behind scaling in one vs two dimensions is never discussed, nor is its significance for understanding spatial representations in these areas.

As the reviewer points out correctly, we refer to 1-dimensional scaling of the object along its elongated dimension. This is the relevant scaling since we characterize the neuron’s spatial selectivity profiles also along this single dimension. To avoid confusion, we removed the previous example about objects appearing in different visual depths (lines 306-308), and now provide a working definition of scale-invariance in our experiment, lines 312-314.

Given these issues it is difficult to grasp the true significance of the results.

We think that the clarifications and additional explanations that we added to the manuscript should make the novelty and relevance of the findings more apparent.

Below are some specific suggestions for improving the presentation.

Introduction

Line 53: I don’t believe ‘subject-independent’ is an adequate synonym for allocentric if that is what this parenthetical statement is implying.

We rephrased the introductory sentence to specify more precisely what we mean and hopefully thereby also make it more accessible to a broad readership.

Lines 56-57: Awkward phrasing, recommend rewriting.

The statement was rephrased (now lines 57-59).

Line 83: Doesn't sound like this result demonstrates strictly allocentric coding, it sounds more context-dependent.

We share the reviewer's view that the reported change of reference frame in this previous study could be a case of context-dependence. To respect the interpretation of the authors and at the same time emphasize the role of context, we changed the sentence as follows (lines 94-97):

"While this points to the possibility of context-dependent allocentric encoding in the visual cortex, the perceptual task of judging object motion during self-motion is very different from aiming a reach towards an object-relative location in terms of spatial-cognitive demand."

Line 118: for clarity suggest rephrasing to something like: '... encode visual cues and reach targets in object-centered as well as egocentric reference frames, with the predominating reference frame in both areas being dynamically adjusted based on cognitive demands'.

Thank you, the statement was rephrased (lines 135-137).

Results

Line 138: for clarify: '... shifted horizontally ...'

Thank you. The change was applied (line 156).

Line 140: for clarity, suggest adding that in this experiment position could be shifted horizontally or vertically.

In Exp II, the reach object always shifted horizontally and vertically relative to the reference object, so that they are always position-incongruent in both horizontal and vertical dimensions. The following sentence was added (lines 158-160):

"In Exp-II, reference and reach object were horizontally and vertically offset to each other, such that they were always position-incongruent in both horizontal and vertical dimensions."

Line 150: would recommend sticking with 'egocentric' rather 'body-centered' to avoid confusion with a 'trunk-fixed' reference frame, which was not strictly examined.

Agree. We changed the "body-centered" to "egocentric".

Line 159: '...encode the locations of visual cues and reach goals ...'

Thank you, change was applied.

Line 165: 'analyses'

Thank you. Corrected.

Line 166-168: See general comments regarding the number of neurons analyzed.

See our answer to the general comment.

Lines 169-172: Although idealized neurons are straightforward, real neurons often show partially shifted and/or scaled response profiles. As a result it is important to point out up front what 'consistent with encoding of cue location in an object centered reference frame' really means. In this example, if the profiles were plotted in object centered coordinates they would likely appear partially shifted and scaled.

The possibility of mixed encoding comes up later but should be discussed prior to presenting real responses and describing them as resembling object-centered or egocentric. Being clear on this point is imperative since the authors later state that they didn't discretely categorize responses as object-centered or egocentric.

We agree. We added a statement earlier in the text, elaborating on the possibility of mixed encoding in these areas (lines 178-183). We also emphasize more the idealized nature of the hypothetical neurons (lines 170-172).

Lines 177-179: I read the sentence beginning with 'Neural response strengths ...' several times and can't decipher its meaning.

We apologize for poor phrasing. We removed the sentence now, as it anyhow was pointing to an irrelevant detail.

Lines 182-184: As indicated above, these statements should come earlier and 'mixed encoding' should be defined.

The statement added earlier in the text now also distinguishes between across-neuron diversity in encoding versus within-neuron hybrid encoding of two reference frames (lines 178-183).

Lines 186-188: Without showing one of these examples it will be very difficult for readers to understand these statements. Also, 'linear selectivity profile'?

We added references to two examples of linear selectivity profiles in Supplementary Fig 3 (lines 213-215).

Lines 191-194: Again, the language here will likely confuse many readers. Don't the authors mean to say that such gradual tendencies 'were quantified to determine whether they differed' rather than 'can be quantified and might differ'?

Thank you. We re-phrased the statement, lines 219-221.

Lines 209-218: It would likely be more helpful to readers to have the all possible scenarios presented first, and in a more fully fleshed manner, followed by a description of the real data. For example, 'recruitment' hypothesis: bimodal distribution with lobes pointing upper right and lower left; 'recoding' hypothesis: unimodal distribution pointing lower right. It might also be useful to entertain the idea of a 'reverse recruitment' scenario (unimodal pointing upper left), if only to emphasize that this non-intuitive scenario is not matched by very many neurons.

We agree. To ease reading, we moved a paragraph from the methods section to here, applied the suggested changes and restructured this section, line 236-246.

Lines 247-249: The description of the statistical analysis here is adequate. Assuming a t-test was used, the exact type of t-test, degrees of freedom, and t-statistic should all be reported.

We added a sentence (line 299) to refer the reader to the methods section (line 752-757), where we describe how significant deviations of RFD values from zero were tested with a permutation test.

Lines 255-257: The 'real world' examples of scaling described here involves scaling in two dimensions and are therefore not good examples of the type of scaling used here, which was along a single dimension. This might seem like nitpicking but readers may wonder about the relevance of the authors chosen scale manipulation so it would be good to justify it with better examples.

We agree. We removed the example we had provided before about objects appearing in different depth, line 306-307, and provided a more detailed working definition of scale-invariance in our experiment, lines 308-312. We also explain that 1-dimensional scaling of object-size fits the approach of 1-dimensional neural selectivity measures used here (lines 312-314).

Lines 267-271: See general comments regarding the number of neurons analyzed.

See our answer to the general comment

Lines 282-284: To avoid confusion, this statement should come earlier, when the experiment is first being discussed.

Thank you for the suggestion. We moved this sentence to the end of the first paragraph of this section, line 320-322.

Lines 284-285: I don't understand what this statement is referring to.

We apologize if this was not understandable. We rephrased the sentence (which moved together with the sentences of the previous comment to line 322-325).

Line 291: Also an inadequate description of a statistical analysis. Please use proper punctuation and exponential notation.

We corrected and unified the statistical reports throughout the manuscript.

Figures

Figure 2b: Suggest labeling these with some identifying information such as cell numbers, areas, etc.

Thank you. Labels were added.

Figure 2c: I found it very difficult to distinguish among the color gradations in this part of the figure which then made it difficult distinguish which line belonged to which object location.

We split the PSTH figure into several panels each one plotting pairs of PSTH from object-left and -right trials, which should be comparable according to object-centered and egocentric hypotheses. We hope that this makes it easier to distinguish different task conditions and to compare the relevant pairs which are most informative about the validity of the two hypotheses.

Figure 3b: I found this plot to be unnecessarily confusing. The authors essentially transformed the data from Cartesian to polar coordinates to compute an angular histogram but retained the

Cartesian coordinates on the axes. Why? Suggest removing the Cartesian axes and just stating that the indices were converted to polar coordinates for this analysis. Incidentally, since they have done this transformation they could use the distance from the origin ('r') to potentially remove neurons that are clearly not biased toward any quadrant (which appear to be many) and are likely influencing their statistical analyses.

Thank you for the suggestion. We removed the Cartesian axes for a cleaner representation. We also added hypothetical figures for this analyses for better understanding.

Figure 3c (caption): Typographical error: 'zerp' should be 'zero'.

Thank you. Corrected.

Figure 4c: As with Figure 2c, the color gradations are too difficult to distinguish.

The color palette was modified.

Discussion

Given the issues raised in the general comments (particularly regarding the frames of reference) it is difficult to comment on this section of the manuscript.

We think that the revised version of the manuscript allows a clearer assessment of the study aims and achievements. We hope that this helps the reviewer in evaluating the significance of the findings.

Reviewer #3 (Remarks to the Author):

The brain needs and employs multiple reference frames to encode the location of objects and events around. Taghizadeh and colleagues' report is a timely and interesting study which show original data which are crucial to understand the duality of egocentric and object-centered representation of action targets. The paper is clear and well written, the study is well designed and the results properly interpreted. Few things are still to be discussed and some points to be better clarified.

Something seems a bit misleading in the title: « ...dynamically adapt to task demand ». Actually the task is the same for all presented trials, that is to identify and remember the target location of an action to be executed at the given signal. What if the goal was different and the subsequent action absent? Would the dynamic of the activation be the same or different during the memory phase? This said, certainly the data show that there's a shift from an object-centered spatial encoding during memory to an egocentric one during planning (beautiful result), but both are elicited by the same task demand at every trial.

Apologies if our wording was misleading. With changing "task demand", we meant the changing cognitive demand over the course of a trial in this task. In response also to reviewer #1, we changed the title to "cognitive demand" and added the following clarification to the introduction, line 77-82:

"This suggests that spatial processing needs to be cognitively controlled so that those spatial parameters are available at different stages of action preparation that are relevant in the respective moment. In the earlier phase, when deciding where along the stick to pick it up, object-centered encoding relative to the stick is most relevant. Later, for planning the physical movement of the hand, egocentric postural signals might be more relevant.."

The object shift manipulation from reference to target position changes the location of the object horizontally, and it seems on the same azimuth on the screen. Why did the authors make this choice? Wouldn't the object-centered reference hypothesis predict similar results even when changing the position of the object vertically from reference to target? Could the constant azimuth itself elicit or at least facilitating the activation shift taken as proxy of the object-centered reference?

It is true that ideal object-centered encoding would be invariant to vertical shifts of the horizontally oriented object also. Yet, the spatial selectivity of the neurons was assessed only along the horizontal dimension in our experiment, due to the linear shape of the object. Therefore, we can predict the to-be-expected selectivity profiles for egocentric encoding only for horizontal shifts of the object position, not for vertical shifts. We need both the object-centered and the egocentric predictions to compare quantitatively which model fits better. If we repeated the same experiment with object-orientation and -shift both being changed to vertical, then we would anticipate the same outcome (mixed selectivity with dynamic shift) as for the horizontal layout.

A further discussion point is which signal triggers the shift from object-centered to egocentric. Unless I'm missing something, no information is given concerning the relative temporal dynamics of the activity in PRR and PMd. Is one leading the other? Are they both receiving signals from elsewhere entailing a shift?

This important question was also raised by reviewer #1. We repeat our answer here.

We agree with the reviewer. We would love to draw strong conclusions about relative latencies between PMd and PRR, but are careful, given the diverse findings between the PI and RFD measure (Fig. 3a/d). Below, at the end of this rebuttal, we will analyze the correlation coefficients (PI measure) in more detail. These results do not support the view that the shift in reference frame occurs first in PMd, as is suggested by the RFD results. We added the following comment to the Discussion section to give a balanced account of this situation (lines 416-423):

“In our decoding analyses, PMd reflected the allo-to-ego transition earlier than PRR (Fig. 3c), while correlation-based measures (Fig. 3a; plus an alternative measure based on the objCorr/egoCorr ratio; data not shown), did not indicate a latency difference between both areas. This means, if there is a latency difference at all, then PMd leads the shift in reference frame. Such finding would support earlier conclusions that frontal areas lead parietal areas in determining motor goals when tasks require a spatial remapping²⁷, and the more general notion that frontal lobe exerts cognitive control over posterior regions of the brain.”

Please see "Relative temporal dynamics between the areas" at the end of this letter for details.

Reviewer #4 (Remarks to the Author):

The paper by Taghizadeh et al. investigates whether single neurons in premotor dorsal and in parietal cortex encode cue and reach target in reaching tasks in object-centered and/or egocentric reference frames of reference. They find that the predominant frame of reference shifts according to cognitive demands in different epochs of the behavioral task. Specifically, in both PMd and PRR the object-centered encoding predominated during visual memory while egocentric encoding predominated during movement planning. The Authors also provide the first

electrophysiology evidence for size-invariant spatial selectivity in both cortical regions. The paper is well written, figures are clear and compelling and data are reliable and analyzed in several ways. I only have a couple of main concerns and a minor one.

Main concerns:

1) Recording site: The Authors report the coordinates of recording chambers. This roughly indicates the cortical region where they record from, but the reader does not know where they exactly record from. For instance, they say they record from PRR, and claimed that “sustained direction-selective neural responses during center-out reach planning (memory period) served as physiological signature to confirm the region of interest”. But neurons active during planning of reaching are present in both SPL and IPL as well as in both walls of the intraparietal sulcus. Where did they actually pick-up the neuronal activity described in the paper? In short, the recording sites must be shown.

We added Supplementary Figure 9, showing the recording sites in detail for both brain areas in both monkeys. Sites were reconstructed based on structural MRI of the brain, including the implanted chambers, and combining them with the within-chamber electrode positions documented during daily recordings.

2) The egocentric frame of reference repeatedly reported in the paper could be an eye-centered, head-centered, or body-centered frame of reference. Although the term ‘egocentric’ used in the paper in comparison with ‘allocentric’ can be accepted, because the used task does not allow to discriminate eye-centered, head-centered, and body-centered frame of reference, in my opinion the point of different egocentric frames of reference should be recalled and discussed in the paper, given that in both parietal and premotor cortex, neurons encoding space in these different frames of reference are reported to be present.

Apologies, if this did not become clear enough in the previous version. To make it more evident, we added explanations to our use of “egocentric” to the introduction (lines 66-68) and methods sections (lines 507-511). Given evidence from previous studies of other labs, we will still refer to the non-object-centered encoding as most likely being egocentric (discussion section, lines 383-387).

Minor concern:

If I’ve correctly understood, the ‘white square’ reported to be used as hand fixation stimulus is shown, in Fig. 1, as a small black square. This raises some confusion. Please modify.

Thank you, we changed the color of the fixation square from black to white.

Relative temporal dynamics between the areas – reference frame vector analysis

The general shift in reference frame between memory and planning period is robust and independent of the exact measure that we used to quantify it. In contrast, the relative timing of the shifts in our data depends on the used measure: The PI (Fig. 3a) and RFD (Fig. 3c) measures of the main manuscript suggest different results regarding relative timing of the reference frame shift from object-centered to egocentric. The PI, based on correlation coefficients, shows no latency differences (Fig. 3a), the decoding approach does (Fig. 3c). This is why we, so far, do not present relative latency as main finding, even though the pattern seen in RFD would nicely complement and support our previous conclusions (Westendorff et al. 2010). For the current data, we had also conducted additional analyses to evaluate potential temporal differences between areas PRR and PMd. Like PI, this method is based on the correlation coefficients. It does not confirm latency differences. We present the results in the following.

For every neuron in every time bin, we plotted the object-centered versus egocentric correlation coefficient, objCorr vs egoCorr. We call the vector that connects every point to the center reference frame vector (RFV; Fig. R-1, green dot/vector). RFVs pointing to below the unity line ($\theta < 45$) indicate spatial selectivity biased towards object-centered encoding and RFVs above the unity line ($\theta > 45$) are biased towards egocentric encoding. The position invariance (PI = objCorr - egoCorr) reported in the main manuscript reflects the horizontal deviation of the data points from the unity line (Fig. R-1, red line). As an alternative, we also analyzed the angle θ of the RFV, which is calculated from the ratio (the arctan of the ratio) rather than the difference of correlation coefficients.

Figure R-1 The reference frame vector (RFV): The green dot shows objCorr and egoCorr of one neuron in one time interval. The vector connecting the center to this dot (the green vector) is the reference frame vector of the neuron. The grey line passing through the center is the unity line. The red line shows the Position Invariance (PI) measure (see Methods for PI).

Fig. R-2 shows the scatter plot of correlation coefficients for all neurons recorded from PMd and PRR in 150 ms time bins sliding by 50 ms around the reach object onset. Small arrows in the plots show the average RFV across neurons. Before presentation of the reach object, the average RFV is pointing below the unity line, indicating object-centered reference frame in both areas. After presentation of the reach object, the average RFV rotates towards egocentric encoding, corresponding to the findings based on PI (and on RFD) reported in the main text.

Figure R-2: Every panel plots objCorr vs egoCorr for all neurons in PMd (top row) and PRR (bottom row) in one time interval. Time intervals are relative to the time of the reach object onset ($t=0$ ms). Spatially, correlation coefficients are calculated relative to the reference object “visual memory”, left panels) or relative to reach object (“movement planning”, right; see Methods and Materials. The arrow in every panel shows average RFVs across neurons.

Fig. R-3 directly compares average RFVs from PMd and PRR. After presentation of the reach object, both vectors start to rotate towards egocentric encoding. It seems that this change initiates first in PMd, since the two vectors do not overlap for the first two time bins ([-150 150] and [-100 200]) of the movement planning and PMd leads the rotation. By the third time bin ([-50 250]), PRR catches up with PMd. This pattern could suggest that the transition to a predominantly object-centered reference frame occurs first in PMd.

Such earlier initiation of the RFV rotation towards egocentric encoding in PMd would be consistent with previous findings that tuning to movement goals occurs first in PMd whenever the task requires spatial remapping (Westendorff et al., 2010). Yet, more detailed analyses questions the support for latency differences in the current data.

Figure R-3 Temporal dynamics of the average RFV. Every panel plots the average RFV across neurons in PMd (red) and PRR (blue) in the polar space for different time intervals relative to reach object onset.

Fig. R-4 provides a representation of the same result split according to angle (Fig. R-4b) and amplitude (Fig. R-4c) of the RFV across 200 bootstrap samples of neurons from either brain area. Both areas reach predominant encoding of egocentric information (crossing of the zero line) around the same time, namely the time window centered on $t=0.25$ ms (Fig. R-4a), i.e., the time bin [100 400] ms (Fig. R-3). The seeming lead of PMd in initiating this rotation (indicated by the difference between RFV in PMd and PRR) in fact is caused by a (paradoxical) brief counter-rotation of the RFV in PRR in the time window [-150 150] ms when calculating correlation coefficients relative to the reach object. Therefore, we cannot conclude that the reference frame change happens first in PMd.

Figure R-4 Amplitude and angle of the RFV: (a) angle of the average RFV in PMd (red) and PRR (blue) for different time intervals around the time of the reach object onset. Time points indicated on the x axis show the center of the time intervals. Shaded error bars show the 95% confidence interval across 200 bootstrap samples of neurons in either area. (b) Difference between the angles of the average RFV (95% CI). (c) Difference between the amplitudes of the average RFV (95% CI).

References

- Avillac, M., Deneve, S., Olivier, E., Pouget, A., Duhamel, J.R., 2005. *Nat.Neurosci.* 8, 941–949.
- Brozovic, M., Gail, A., Andersen, R.A., 2007. *J.Neurosci.* 27, 10588–10596.
- Deneve, S., Latham, P.E., Pouget, A., 2001. *Nat.Neurosci.* 4, 826–831.
- Deneve, S., Pouget, A., 2003. *Neuron* 37, 347–359.
- Westendorff, S., Klaes, C., Gail, A., 2010. *J.Neurosci.* 30, 5426–5436.

REVIEWER COMMENTS

Reviewer #1 (Remarks to the Author):

My overall impression was already very positive last time and I mainly had minor comments regarding the clarification of definitions and method details and further discussion of the results with regard to recent findings in the literature. The authors satisfactorily addressed all my comments. I have no further points to add.

Reviewer #2 (Remarks to the Author):

The authors have made several changes to the text and figures that greatly improve the readability of the manuscript. However, although some attempts have been made to improve the presentation of the statistics, these descriptions are still ambiguous in several instances, as described below.

First, regarding the issue of the relatively low number of neurons recorded, particularly in Exp-II, the authors assert in their response to the reviewers that they only reported results that were 'consistent across both animals' and that they found effects 'across different animals at the neural population level'. However, there is nothing in the main manuscript (aside from some single cell examples) that would allow a reader to independently assess those assertions.

Figure 5 shows PSI and RFD data for the populations, but these were generated using combined data (neurons) from both animals. The plots show that the distributions are biased toward positive values and the statistics reported in the caption for the PSI data (based on a Wilcoxon signed rank test) support that. Notably the results point to p-values that are less than 10^{-9} , which is a bit surprising given the shape of these distributions (see below).

Supplementary Figure 7 does show separate PSI data for the two animals but these plots quite clearly indicate that the data from the two animals are not very consistent. The data for monkey K are strongly biased toward positive values while monkey H's are not. Statistics are not separately reported for the two animals but it wouldn't be surprising if monkey H's data were determined to be unbiased when variability is taken into account.

I do not believe that this is nitpicking. The authors are attempting to make strong statements regarding the functions of these areas in monkeys, and (presumably) by analogy humans, based on a handful of neural recordings from two animals. It is incumbent upon them therefore to be as straightforward as possible about the limitations of their data and analyses. Regarding statistical analysis of neural data, it is important to point out that some have noted that samples (neurons) obtained from the same animal are not technically independent, which violates one of the key assumptions behind most hypothesis tests, including non-parametric ones like the signed rank test (e.g. Aarts et al., 2014). All the more reason to be careful in asserting too much based on limited data.

As an aside, no statistics are reported for the RFD data in the caption of Fig. 5. P-values are reported in the associated main text (line 357 – should this reference say 5c, not Fig 4c?) and the methods refer to randomization tests, though the description is inadequate. Incidentally, why was a signed rank test used for one measure (PSI) and a permutation test for the other (RFD)? The authors need to justify these different approaches.

In the end, I think these data are interesting and I appreciate the difficulty involved in acquiring and analyzing them. However, transparency is always the best policy. If the numbers are low and the results of statistical analyses differ between animals, just say so. It could always be said that despite that, the data were trending in the same direction for both animals and let the reader decide how to interpret the discrepancy.

Papers cited in this review:

Aarts et al., Nature Neuroscience, vol 17, number 4, 2014

Reviewer #4 (Remarks to the Author):

The authors answered all my questions satisfactorily. I have not further comments.

Patriia Fattori

We thank all the reviewers for critically reassessing our manuscript and are happy to learn that most of the concerns could be resolved with our previous revision. We highly appreciate the careful review of reviewer #2 that gives us the opportunity to clarify some remaining points about the statistics, which we agree are important to be cleared.

We added detail to our previous statistics, namely splitting the data of Exp-II for the two animals, and show time-resolved results of our existing PSI measure. We also include additional analyses (demixed PCA) on the same question; originally, we had decided against including a third measure not to overload the paper. These additional analyses confirm our previous assessment that in Exp-II each animal individually shows results supporting our conclusions, despite the fact that the database for animal H is smaller, and results accordingly are less robust than for K. We added the new analyses for Exp-II to the supplement, since we feel that they strengthen the paper.

We will explain details in the point-by-point rebuttal below.

REVIEWER COMMENTS

Reviewer #2 (Remarks to the Author):

The authors have made several changes to the text and figures that greatly improve the readability of the manuscript. However, although some attempts have been made to improve the presentation of the statistics, these descriptions are still ambiguous in several instances, as described below.

First, regarding the issue of the relatively low number of neurons recorded, particularly in Exp-II, the authors assert in their response to the reviewers that they only reported results that were 'consistent across both animals' and that they found effects 'across different animals at the neural population level'. However, there is nothing in the main manuscript (aside from some single cell examples) that would allow a reader to independently assess those assertions.

We thank the reviewer for catching this; it was unintentional. We meant to report the test results for the individual animals together with the supplemental figure that we had added. Below we explain how we tested for consistency between both animals and added this information to the manuscript.

Figure 5 shows PSI and RFD data for the populations, but these were generated using combined data (neurons) from both animals. The plots show that the distributions are biased toward positive values and the statistics reported in the caption for the PSI data (based on a Wilcoxon signed rank test) support that. Notably the results point to p-values that are less than 10^{-9} , which is a bit surprising given the shape of these distributions (see below).

We double-checked the signed rank test. We can confirm that the reported p-values match the data shown. As sanity check, we simulated random data consisting of $N=70$ data points (number of neurons in PRR) either as a uniform or a normal distribution. We shifted the data such that 12 neurons would fall below zero (this was the observed number in the empirical data) and applied the signed rank test. p-values for the normal distribution are in the order of magnitude of 10^{-9} and even smaller for the uniform distribution.

For the permutation test that is used for the RFD data, the estimated p-values are extremely small and can vary by 1 or 2 orders of magnitude depending on the random numbers. To avoid such reports, we decided to follow the common practice suggested by a previous reviewer and

for all test results only report $p < 0.001$ as the minimum and the exact p -values if they are larger than 0.001.

Supplementary Figure 7 does show separate PSI data for the two animals but these plots quite clearly indicate that the data from the two animals are not very consistent. The data for monkey K are strongly biased toward positive values while monkey H's are not. Statistics are not separately reported for the two animals but it wouldn't be surprising if monkey H's data were determined to be unbiased when variability is taken into account.

We added the test results for the individual animals to the supplemental figure (we assume Figure 8 must have been meant). We also added a paragraph (line 351) about similarities and differences between monkeys with reference to the Supplementary Figures 8 and 9 containing more detailed analyses.

The PSI distribution in Supplementary Fig. 8 show a significant positive shift for monkey K (signed rank test, PRR and PMd $p < 10^{-7}$). Monkey H, while showing a non-significant positive trend only in this time window, shows stronger position and size invariant encoding earlier in the memory period. The following figure shows the PSI, in 300 ms time windows of the memory period, sliding by 50 ms. The green circles above the box plots indicate significant shift of the distributions (signed rank test $p < 0.05$), and purple asterisks indicate significance after a correction for multiple comparisons across time intervals using false discovery rate correction. This is a very conservative correction here since consecutive time windows are statistically not independent due to their overlap. From signed rank test on every time bin, for monkey K all bins show $p < 10^{-7}$, monkey H in the significant time bins $p < 0.034$. We added this explanation to the caption of the Supplementary Fig. 8.

The RFD in late memory period had a significant positive shift for monkey K in both areas (RFD>0, PRR and PMd $p < 0.001$), for monkey H, although the distribution is trending towards positive values, the test results was not significant.

The observation that each of our animals individually supports the conclusions is also based on population-level analyses which we originally had not planned to elaborate on in the paper, for simplicity. We conducted dimensionality reduction analyses (demixed principle component analysis, dPCA) on the neural dynamics to test for position and size invariance encoding. The dPCA results show that in both monkeys the best clustering of spatial memory information is achieved when considering an object-relative frame of reference, not an egocentric frame of reference. This analysis is now added to the supplementary information under 'Demixed principal component nalysis of ExplI' and Supplementary Fig. 9.

The authors are attempting to make strong statements regarding the functions of these areas in monkeys, and (presumably) by analogy humans, based on a handful of neural recordings from two animals. It is incumbent upon them therefore to be as straightforward as possible about the limitations of their data and analyses. Regarding statistical analysis of neural data, it is important to point out that some have noted that samples (neurons) obtained from the same animal are not technically independent, which violates one of the key assumptions behind most hypothesis tests, including non-parametric ones like the signed rank test (e.g. Aarts et al., 2014). All the more reason to be careful in asserting too much based on limited data.

We are confident that our claims are well supported by our data and that we do not draw exaggerated conclusions. About Exp-II we say in the Discussion section:

“object-centered encoding was accompanied with size-invariant spatial encoding of different on-the-

object sites, at the single neuron and population levels. To our knowledge, this is the first electrophysiology evidence showing that the fronto-parietal network utilizes size invariant positional code for movement planning [...] PRR and PMd are along the course of the dorsal visual processing pathway and we showed that they express allocentric and size-invariant encoding with a short latency of a few hundred milliseconds after presentation of an object-relative spatial cue”.

This means, we point to the existence of a certain form of encoding in two specific areas of rhesus monkey brains, which had not been reported before. We do not make any inference on other species, nor do we claim exclusive encoding of this type in these brain areas. The significant majorities of neurons supporting this form of encoding among the recorded 70/66 (PRR/PMd) neurons is more than “a handful” and, correspondingly, reported measures indicate highly significant results. To prove the existence of a certain form of encoding, even less than the majority of neurons would be sufficient; we speak of “predominance” in other parts of the manuscript when referring to the fact that the population statistics show a significant shift in favor of the reported encoding scheme.

The neurons were recorded with individually movable, single-contact electrodes, lowered into the cortex on a daily basis. There is no risk of sampling the same neuron on multiple channels or the same neuron repeatedly. Hence, the data presented here is different from chronically implanted array recordings for which sometimes subsequent recordings are reported in the literature as independent measures with nominally very high neuron counts. Such counts would indeed harbor the risk of being inflated unless there is evidence that neurons are different from day to day. The response patterns of our neurons are highly idiosyncratic (see examples in the supplement). Therefore, we can and need to treat them as statistically independent and do not see a reason why the statistical test, esp. the ones performed individually within each animal, should not be valid.

As an aside, no statistics are reported for the RFD data in the caption of Fig. 5. P-values are reported in the associated main text (line 357 – should this reference say 5c, not Fig 4c?) and the methods refer to randomization tests, though the description is inadequate. Incidentally, why was a signed rank test used for one measure (PSI) and a permutation test for the other (RFD)? The authors need to justify these different approaches.

Thank you for catching this, we fixed both, the erroneous figure reference and lacking p-values in the figure caption. We also improved the description of the randomization test in Methods to make the procedures clearer (line 723-735).

Whenever we computed measures on a single neuron basis and compared them between conditions, we used non-parametric testing, with the N being determined by the sample size of the neurons. We used non-parametric tests, because we wanted to be conservative with regard to potentially non-normal distributions.

For the decoding approach, instead, the number of neurons is not a relevant number, since only a single performance value is obtained for the whole population of neurons per each calculation. Since decoding requires the definition of subsamples of trials to define training and test data, respectively, to compute decoder performance, the process has an inherently random component. Randomization tests therefore are an adequate and common way of accounting for the unavoidable inherent randomness. They make the performance estimate independent of the specific randomly drawn training and test sets of a single run. Yet, the $N_{\text{permutation}}$ decoding performance values that result from the number of iterations of the randomization process are not a valid sample for an inference test (e.g. signed rank test), since the N of this sample is an arbitrary choice. Instead, the p-value is determined by the probability of the decoding performance to be smaller than zero; this probability can be directly be measured from the distribution of performance values.

In the end, I think these data are interesting and I appreciate the difficulty involved in acquiring and analyzing them. However, transparency is always the best policy. If the numbers are low and the results of statistical analyses differ between animals, just say so. It could always be said that despite that, the data were trending in the same direction for both animals and let the reader decide how to interpret the discrepancy.

We agree to the importance of transparency. We provide the methodological details and sample sizes (and the data and scripts) to allow the readers their own assessment. Each animal individually shows significant effects in the dPCA and also in the PSI measure (in PSI in time bins of the delay period that differ between animals) and trends in other measures, which we show. Since limited statistical power would be more of a problem for null findings than for observed significant effects, we are confident that our findings are robust and well supported by the data.

REVIEWERS' COMMENTS

Reviewer #2 (Remarks to the Author):

The revisions made to the manuscript that are reflected in the current submission are sufficient.